# DynaVieW: Schema-Guided World Modeling for Understanding Hierarchical Visual Dynamics

Silin Gao [1] [*]   Hao Zhao [1] [*]   Zeming Chen [1]   Sepideh Mamooler [1]   Antara Raaghavi Bhattacharya [1] [2]
Qiyu Wu [3]   Hiromi Wakaki [3]   Yuki Mitsufuji [3]   Li Mi [1] [4]   Syrielle Montariol [1]   Antoine Bosselut [1]

## Abstract

Multimodal LLMs struggle to systematically model the temporal evolution of visual scenes in videos or multi-image sequences. Such inputs require models to predict or simulate multiple levels of dynamic constituents, such as actions taken in the visual sequence, and the associated changes to the visual environment that result. To address this challenge, we propose a dynamic schema-guided world model, DynaVieW, optimized for visual dynamic prediction and simulation. DynaVieW achieves an in-depth understanding of visual dynamics by learning interleaved state-transition sequences, where states cover broad visual scenes from video keyframes, and transitions capture comprehensive dynamic constituents within a hierarchical schema. DynaVieW jointly models transition prediction and state simulation under a mixture-of-experts architecture, with a cross-expert selective attention and a schema token reweighted loss, to ensure effective and robust learning. DynaVieW's superior visual dynamic understanding boosts its downstream performances on both visual narrative creation and world simulation, showing improved consistency and controllability of visual generation and better instruction-following ability.[1]

## 1. Introduction

Large language models (LLMs) are trained for increasingly advanced foundational capabilities (Grattafiori et al., 2024;

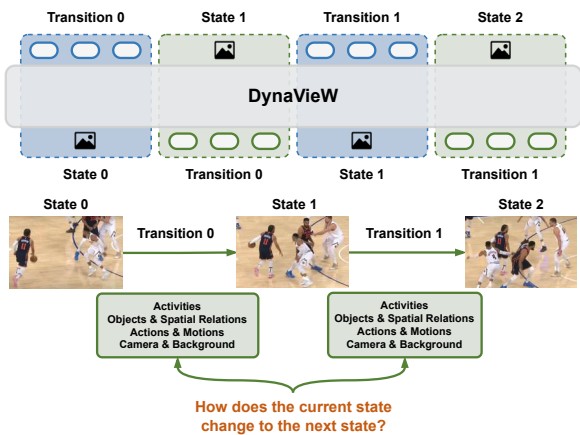

*Figure 1.* **Overview of DynaVieW.** We formulate our world modeling task as learning interleaved state-transition sequences. In particular, DynaVieW jointly learns the prediction of hierarchical transition descriptions (blue) and the simulation of visual states (green) in an alternating manner.

Singh et al., 2025; Comanici et al., 2025; Yang et al., 2025). No longer purely limited to natural language processing, they perform multimodal understanding and generation, enabling abilities such as world perception and simulation (Qin et al., 2024; Duan et al., 2025), and tasks such as visual narrative creation (Gao et al., 2025; Zhuang et al., 2025).

Despite their improvements on static multimodal tasks (*e.g.*, image captioning), multimodal LLMs still struggle to understand *visual dynamics* — the changes that occur in visual scenes as actions are carried out. For example, a visual scene of *people playing basketball* consists of various *dribbling* and *passing* actions, which further include fine-grained *body movements* and *changes of spatial relations between players*, as shown in Figure 1. Simulating such a scene requires multimodal LLMs to understand these multi-level *constituents* (e.g., high-level events such as actions, low-level transformations in the scene linked to these actions) and how they change the progression of the scene. When multimodal LLMs fail to learn such visual dynamics, they suffer from reliability and controllability issues, *e.g.*, they may generate images with poor alignment to input prompts (Wu

*Equal contribution [1]EPFL, Switzerland [2]Harvard University, United States [3]Sony Group Corporation, Japan [4]ETH Zurich, Switzerland. Correspondence to: Silin Gao <silin.gao@epfl.ch>, Hao Zhao <hao.zhao@epfl.ch>, Antoine Bosselut .

*Proceedings of the 43rd International Conference on Machine Learning*, Seoul, South Korea. PMLR 306, 2026. Copyright 2026 by the author(s).

[1]We release our data and code to the community, our project GitHub: https://github.com/Silin159/DynaVieW

et al., 2024), or demonstrate inconsistencies when generating longer-form visual sequences (Gao et al., 2025; Zhuang et al., 2025).

To improve visual dynamic understanding and generation, prior work pre-trains multimodal LLMs on interleaved vision-text data (Alayrac et al., 2022; Zhu et al., 2023). Interleaved vision-text pre-training uses natural sequences of visual content, such as storyboard images or video keyframes, which natively contain rich visual dynamics. In these datasets, the text between visual inputs enables in-place explanations of each step or key point in the progression of the visual narrative, which promotes the understanding of visual dynamics and more precise control of visual sequence simulation. Despite these datasets, most prior interleaved vision-language LMs still fall short of systematically learning visual dynamics, with many approaches (Tian et al., 2024; Chern et al., 2024; Lin et al., 2024a) only exposed to visual contents in pre-training data that have weak alignment between the visual and textual components of interleaved sequences. Other methods were trained on only a single type of visual dynamics, such as motion and scene shifts (Deng et al., 2025), robot-view actions (Qu et al., 2025), or game-view interactions (Zhang et al., 2025; He et al., 2025).

In this paper, we elevate the world modeling capabilities of interleaved vision-language LMs via continued pre-training on broader and more comprehensive descriptions of visual dynamics. Our proposed **Dyna**mic **Vi**sual Sche**ma**-guided **W**orld model (**DynaVieW**) is pre-trained on interleaved state-transition sequences, as shown in Figure 1, across broad domains. Specifically, DynaVieW learns to simulate visual state sequences that are sourced from keyframes of diverse real-world videos, covering various human daily activities, robotic manipulations, art works and auto-driving recordings. DynaVieW learns to predict *transitions* between visual states, represented as text formatted in a hierarchical JSON schema, to comprehensively capture both high-level progression of activities and low-level changes of visual details in a structured manner. DynaVieW adopts a mixture-of-Transformer-experts (MoT) architecture (Deng et al., 2025) to unify the modeling of transition prediction and state simulation. The MoT experts feature a shared multimodal selective attention, which facilitates robust learning of the long state-transition sequences, by dropping out redundant historical information. Moreover, DynaVieW uses a schema token re-weighted cross-entropy (CE) loss to balance its learning of the transition schema format and more specific slot values filled into the schema.

Our DynaVieW model, trained on our broad state-transition data, builds more in-depth understanding of visual dynamics in the world, enabling greater controllability on visual sequence generation by verbalizing hierarchically-defined transition characteristics. In a visual narrative (Gao et al., 2025) task, narrative images generated by DynaVieW achieve significantly better consistency and overall quality compared to baseline models, no matter whether prompted for zero-shot generalization or fine-tuned on task-specific data. We further test the controllability of visual narrative, by prompting the model with the same high-level storyline but different low-level scene descriptions (indicating desired visual details), and find that DynaVieW's outputs can more flexibly adapt to varied input prompts. Finally, we show DynaVieW outperforms baseline models on world simulation tasks (Lai et al., 2024), showing better instruction following in more deterministic visual generation scenarios.

## 2. World Modeling on Hierarchical Dynamics

We formulate our DynaVieW modeling task as learning interleaved state-transition sequences, as shown in Figure 1. The states in each sequence are images depicting a dynamic visual scene, such as shots of *a person playing the keyboard* in Figure 2, and the transitions are JSON-schema texts hierarchically describing changes between adjacent states, such as *the player's right hand moving down to touch the keyboard*. DynaVieW jointly learns the prediction of each transition and the simulation of each state (excluding the first initial state) based on preceding states and transitions, which fosters in-depth understanding of visual dynamics.

### 2.1. DynaVieW State-Transition Data Construction

Figure 2 presents an overview of how we construct the interleaved state-transition data for training our DynaVieW model. We describe the data construction pipeline below.

**State Extraction**    Noticing that a rich amount of human world activities or scenes have been recorded in videos, which naturally illustrate various visual dynamics, we extract video keyframes to serve as the visual states in our DynaVieW learning data. To cover visual dynamics across broad domains, we select a diverse collection of source videos to extract the states, including: (a) Ego4D (Grauman et al., 2022) videos that record ego-centric human daily life activities in household, workplace, *etc.*, (b) AgiBotWorld-Alpha (Bu et al., 2025) videos that contain robot-view trajectories in environments such as restaurant, office, *etc.*, (c) ShareGPT4Video (Chen et al., 2024a) that collects various real-world videos such as cooking tutorials, auto-driving recordings and aesthetically appealing stock videos.[2]

We follow ShareGPT4Video (Chen et al., 2024a) to extract keyframes from above videos in two steps. First, we select candidate keyframes of each video by down-sampling its frames at about 1-second intervals. To avoid selecting blurred candidates in the course of the video, our down-

---

[2]We exclude the Ego4D portion of videos collected in ShareGPT4Video to avoid duplication.

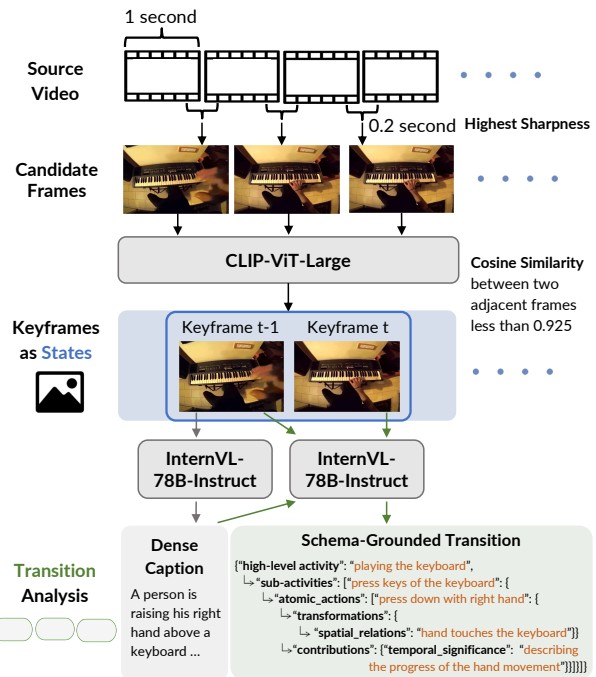

*Figure 2.* **Overview of DynaVieW data construction.** We extract video keyframes as our visual states, ensuring high image sharpness and low CLIP embedding similarity across selected keyframes. Based on that, we prompt an oracle VLM (InternVL-78B-Instruct) to analyze the transitions between adjacent states.

sampling interval is not fixed as exactly 1 second, but allows a $\pm 0.1$-second window of adjustment, as shown in Figure 2. We traverse all frames within each $1 \pm 0.1$-second interval window and select the frame with the highest sharpness measured by Laplacian variance (Laparra et al., 2016). We then traverse the sequence of sampled candidates, and select the final keyframes by keeping only the candidate whose content is significantly different from its preceding selected keyframe.[3] Specifically, as shown in Figure 2, we calculate the cosine similarity of CLIP (Radford et al., 2021) embeddings of the candidate and its preceding selected keyframe, and select the candidate as a keyframe if the similarity is lower than a threshold (0.925).

In Table 5 (Appendix A.1), we present detailed statistics of the source videos and extracted visual states (keyframes) that are included in the DynaVieW pre-training data. The extracted visual state sequences cover diverse lengths (from 2 to 100 states per video) and content, due to the diversity of the original source videos. Our state extraction is also monitored by the sharpness measure and CLIP embedding similarity, which ensures the image quality and allows flexible time intervals (from 0.8 to 167.1 seconds) between adjacent states. The flexibility of time intervals enables DynaVieW

to learn both **short-term** and **long-term** visual dynamics, such as *a person's rapid movement in a few seconds* and *a long-term monitoring of a sunset*, respectively.

**Transition Frame and Annotation** For each extracted sequence of visual states, we then annotate the transition between each pair of adjacent states in the sequence. To build hierarchical descriptions of visual dynamics, we frame the transition based on a fine-grained JSON schema, as illustrated in Figure 2. Our transition schema captures both the progress of **high-level** activities (or events) and the changes of **low-level** visual details, with analysis of how the low-level changes contribute to the high-level progress.

**(a) High-Level Activity** describes the overarching scenario occurring between the adjacent states, *i.e.,* a general description of the primary action or event taking place; **(b) Sub-Activities** are constituents of the high-level activity, such as intermediate-level actions or events that contribute to the overall activity; **(c) Atomic Actions** break down each sub-activity into the smallest meaningful units that cannot be further decomposed while maintaining semantic meaning.

Each atomic action can result in one or more of following transformations when transitioning between the adjacent states: **(d.1) Object Transformations** capture whether an object is newly introduced into, removed from or kept persistent in the scene; **(d.2) Object State Transformations** detect whether the color, shape, physical status or texture of any object is changed; **(d.3) Spatial Relation Transformations** describe whether the positional relationship, contact or alignment of objects is altered; **(d.4) Action Transformations** capture whether any continuation, completion, initiation or interruption of action has occurred; **(d.5) Motion Transformations** detect whether any translational, rotational, oscillatory or deformation motion has happened; **(d.6) Camera Transformations** describe whether the camera has any viewpoint changes, zoom operations, pan or tilt movements, or focus adjustments; **(d.7) Background Transformations** detect whether the background has experienced any scene shifts, or changes of illumination, atmospheric conditions or contextual elements.

We analyze the overall contributions of above transformations from three aspects: **(e.1) Atomic Action Contribution** describes how the transformations enable, facilitate, or result from the atomic action; **(e.2) Sub-Activity Contribution** analyzes how the transformations support or advance the broader sub-activity goal; **(e.3) Temporal Significance** captures the timing and sequence importance of the transformations within the overall high-level activity flow.[4]

Based on above schema, we prompt InternVL-78B-Instruct (Chen et al., 2024b) to annotate each state transition in two

---

[3]The first candidate is always selected as a final keyframe.

[4]We present a complete example of our state transition schema in Figure 5 in Appendix A.1.

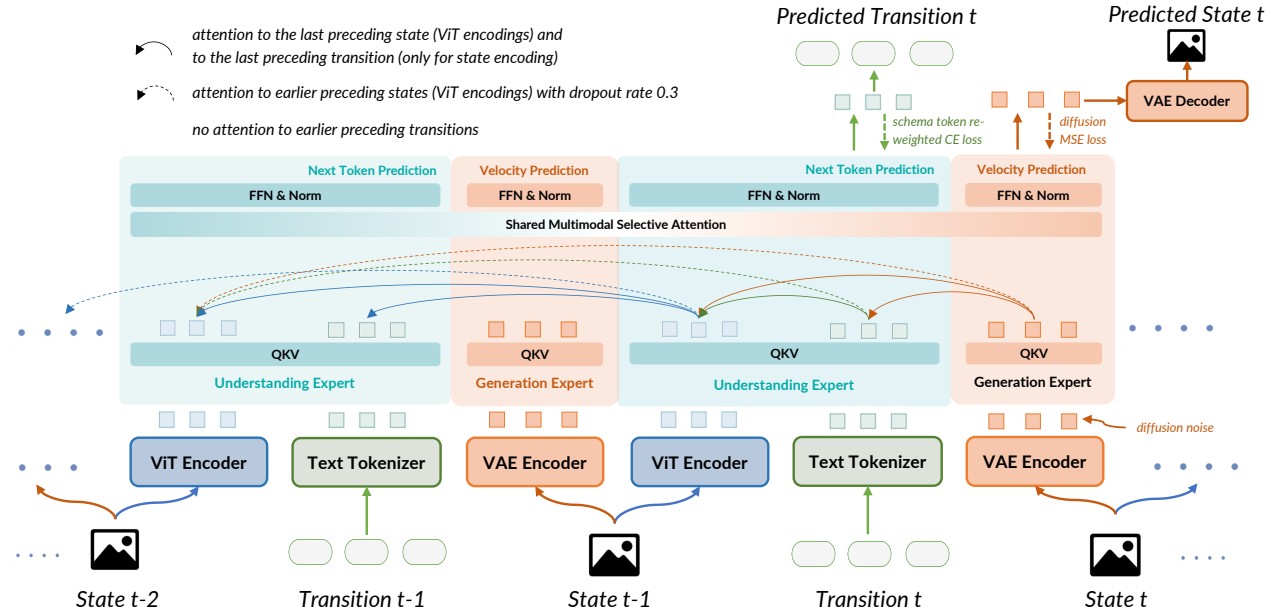

*Figure 3.* **Overview of DynaVieW modeling approach.** The architecture of DynaVieW adopts a mixture of world understanding and generation experts, with shared multimodal selective attention. DynaVieW is jointly trained on the transition prediction and the state simulation, with a schema token re-weighted cross-entropy (CE) loss and a diffusion mean squared error (MSE) loss, respectively.

steps, as shown in Figure 2. Given a pair of adjacent states, we first prompt InternVL to generate a dense caption of the former state, which guides the model to understand the events, actions and visual details shown at the start of the state transition. On top of the dense caption, we then instruct InternVL to compare the latter state with the former state, and generate transition descriptions following our defined schema. To ensure that the output of InternVL is in valid JSON structure, we use constrained decoding (Hokamp & Liu, 2017) to restrict the token generation.[5]

Our hierarchical transition descriptions enable DynaVieW to learn a comprehensive understanding of how human world events or actions and their associated visual elements are coherently arranged and transformed over time. Moreover, the fine-grained verbalization of transitions also facilitates DynaVieW's intermediate thinking and planning of the latter visual state simulation, which contributes to better controllability of visual generation. These promote DynaVieW to build a more general and reliable world modeling.

### 2.2. DynaVieW Modeling Approach

Figure 3 shows the overall architecture of DynaVieW. We adopt a mixture-of-Transformer-experts (MoT) architecture following BAGEL (Deng et al., 2025), to integrate both world understanding and generation capabilities into Dy-

naVieW. For world understanding, each state that provides contexts for predicting subsequent transitions and states is encoded by a vision Transformer (ViT), using SigLIP2 (so400m-patch14-384; Tschannen et al., 2025) architecture with NaViT (Dehghani et al., 2023) approach of preserving the native image aspect ratio. The ViT encodings of state tokens (or patches) and the embeddings of transition tokens are both routed to an understanding expert with Qwen2.5 (7B; Qwen et al., 2025) LLM architecture. In terms of world generation, a variational autoencoder (VAE) FLUX (1-dev; Labs, 2024; Labs et al., 2025) is used to encode and decode each state that needs to be predicted. The VAE encodings of state tokens, after corrupted with diffusion noise for rectified-flow training[6], are routed to a generation expert with the same Qwen2.5-7B architecture. The understanding and generation experts perform shared multimodal self-attention at every Transformer layer.

**Multimodal Selective Attention** We employ a selective mask in the multimodal self-attention of DynaVieW, as shown in Figure 3, to reduce redundant information and avoid naive failure modes in the state-transition sequence modeling. Specifically, we mask out each transition's attention to its preceding transitions, and only keep the attention to preceding states' ViT encodings that implicitly reveal the

---

[5]We include our JSON schema used for constrained decoding and prompts for transition annotation in Appendix A.1.

[6]To save the token budget of DynaVieW and thus enable longer state-transition sequence modeling, we exclude the noise-free VAE encodings used as additional decoding conditions in BAGEL, which have minor impact on improving DynaVieW learning.

historical transitions. This effectively prevents DynaVieW from naively copying one of the historical transitions as the prediction. Similarly, we only keep each (either ViT or VAE encoded) state's attention to the last transition that leads to the state, and attention to its preceding states' ViT encodings as contexts, while mask out the attention to earlier transitions, to avoid them interfering with the state simulation. In order to mitigate potential overfitting to our training data distribution, we also employ a dropout of attention to the preceding states. In particular, for each state (or transition), we always keep its attention to the last state preceding it, which contains the most recent visual contexts for prediction, but mask out the attention to each of the earlier states with a probability of 0.3. Besides, we adopt causal attention within each transition's text tokens, and bidirectional attention within each state's ViT or VAE tokens.

**Training Objectives** We initialize DynaVieW with the pre-trained weights of BAGEL model checkpoint (BAGEL-7B-MoT), and continue to train DynaVieW on both transition prediction and state simulation tasks. The two tasks are performed alternately in an interleaved sequence, as shown in Figure 3. DynaVieW performs next token prediction on top of the understanding expert to auto-regressively generate each transition, which is trained on the cross-entropy (CE) loss. Note that our defined transition consists of JSON-schema tokens with only slight variations across different training samples, and slot-filling tokens that can vary dramatically across different samples, as illustrated by the **black** and orange tokens in Figure 2, respectively. To prevent DynaVieW from overfitting to the relatively static (and thus more frequently appeared) JSON-schema tokens, while under-learning the more dynamically varied slot-filling tokens, we adopt a **schema token re-weighted CE loss** for training the transition token prediction. Specifically, we parse each gold reference transition to identify its JSON-schema tokens, and set a lower CE loss weight (0.1) on these identified tokens, while keep a normal CE loss weight (1.0) on the rest of slot-filling tokens. This balances DynaVieW's learning of the transition structure and the more specific transition details. For state simulation, DynaVieW performs velocity prediction (Salimans & Ho, 2022) on top of the generation expert, to simulate the de-noised VAE encodings of each state image. The predicted de-noised encodings are then compared to the gold state's clean VAE encodings, to calculate the mean squared error (MSE) loss for diffusion training. At the inference phase, pure Gaussian noise is sampled as the input to the generation expert for each state simulation, and the VAE decoder is used to generate the simulated state image based on the predicted de-noised encodings. We sum the CE loss for transition prediction and the MSE loss for state simulation with equal weights (1.0) as DynaVieW's final training loss.[7]

## 2.3. DynaVieW Validation

In this section, we validate DynaVieW's performance of world modeling and the quality of our state-transition data construction. Specifically, we sample 900 more source videos, 300 each from Ego4D, AgiBotWorld-Alpha and ShareGPT4Video datasets (excluding videos in the training data), and construct their state-transition data using the pipeline described in Section 2.1. Based on that, we use GPT-4o (Hurst et al., 2024; Achiam et al., 2023) as a judge to evaluate DynaVieW's predicted transitions and simulated states on these 900 validation samples, and meanwhile to check the gold references constructed by our pipeline.

For each predicted (or gold) transition, we prompt the judge to check the filled values in each type of the transition slots defined in our JSON schema, including three levels of **Activities** (high-level activity, sub-activities and atomic actions), seven types of **Transformations** (objects, object states, spatial relations, actions, motion, camera and background), and three types of **Contributions** (to atomic action, to sub-activity and temporal significance). The judge evaluates whether (Accept) or not (Reject) each type of slot filling values describes a reasonable prediction or analysis that accords with spatial-temporal causality and commonsense, given the transition's preceding states as contexts. A "None" result is given if a type of slots is empty with no filled value. For each simulated (or gold) sequence of states, we instruct the judge to first score the **Style Consistency** of the states on a Likert scale from 1 to 10 (higher is better). Similar to the evaluation of Transformations, the judge then evaluates each state in the sequence, *w.r.t.* whether (Accept) or not (Reject) each type of the state's **Depicted Transformations** on top of its preceding states are aligned with spatial-temporal causality and commonsense. A "None" result is also given if a type of Transformations is not depicted in the state.[8]

Table 1 shows our validation results on DynaVieW outputs and gold references. DynaVieW predicts transitions with fairly low Reject rates on all three aspects, which are not exceeding 2% and comparable to the gold annotations. This indicates that the transitions predicted by DynaVieW are reliable and the annotated gold transitions are also accurate. For state simulation, DynaVieW also achieves a high style consistency score (above 8.5 out of 10, comparable to gold states) and a low Reject rate (less than 5%) of depicted transformations, indicating a decent capability of visual dynamic sequence generation. Although the precision of DynaVieW's simulated transformations slightly falls behind gold references, with ∼ 2% higher Reject rate, it generates richer transformations as a trade-off for higher recall, with higher Accept rate and lower None rate. Our validation results verify the effective learning of DynaVieW based on

---

[7]More details of DynaVieW's training and inference methods

are presented in Appendix A.2.

[8]We include our judge's validation prompts in Appendix A.3.

*Table 1.* Validation of DynaVieW transition prediction and state simulation, compared to the gold references, using GPT-4o as the judge.

| Method | Transition Prediction | | | | | | | State Simulation | | | |
|--------|-----------------------|---|---|---|---|---|---|------------------|---|---|---|
| | Activities | | Transformations | | | Contributions | | Style Consistency | Depicted Transformations | | |
| | Accept (%) | Reject (%) | Accept (%) | Reject (%) | None (%) | Accept (%) | Reject (%) | Score (1-10) | Accept (%) | Reject (%) | None (%) |
| DynaVieW | 98.11 | 1.89 | 43.57 | 0.53 | 55.90 | 98.00 | 2.00 | 8.54 | 65.39 | 3.48 | 31.13 |
| Gold | 98.89 | 1.11 | 44.19 | 0.43 | 55.38 | 98.67 | 1.33 | 8.89 | 58.89 | 1.22 | 39.89 |

our high-quality state-transition data.

## 3. Experimental Methods

The comprehensive learning of human world visual dynamics fundamentally boosts DynaVieW's better adaptation to various downstream multimodal tasks. We test the advantages of DynaVieW on several downstream benchmarks that focus on two complementary capabilities: visual narrative generation (Gao et al., 2025) and instruction following in world simulation (Lai et al., 2024).

**Visual Narrative Generation** Our primary evaluation targets the visual narrative generation task, *i.e.*, creating continuations of an image sequence to depict the scenes of a narrative, which requires understanding narrative-driven state transitions, long-horizon consistency across narrative images, and progression of causal events. We adopt the Visual Writing Prompts (VWP) portion of VinaBench (Gao et al., 2025) as our evaluation benchmark, which covers diverse dynamic scenes and narrative characters from movies. Each VinaBench sample consists of a sequence of images depicting the visual narrative scenes, where each image is aligned with a textual scene description and a constraint capturing key elements (such as time, character and location) in the image and connections between visual and textual entities (such as linking characters appeared in the image to their names mentioned in the text).

Given an initial narrative image (or scene) and a textual description of the next scene, the model should generate a realistic and coherent continuation image to depict the next scene. The generated image and its description are appended to the context, enabling iterative prediction along the full narrative trajectory. We test the model performances under both zero-shot generalization and supervised fine-tuning (SFT). In the SFT setting, the model is fine-tuned on VinaBench training samples to also predict the constraint of the next scene, which serves as additional thinking process before simulating the next scene image. Based on the constraint of each scene, we follow VinaBench to evaluate the model on two aspects: per-scene text-image alignment and cross-scene consistency. We employ powerful oracle LLMs, GPT-4o (Hurst et al., 2024) and Gemini-2.5-Pro (Comanici et al., 2025), as the judges to measure alignment and consistency, based on VQAScore (Lin et al., 2024b) and judgment

questions *w.r.t.* the constraint.[9]

**Instruction Following in World Simulation** In contrast to long-horizon and open-ended visual narrative generation, where multiple scene continuations may be valid, world simulation requires translating textual instruction (or action) into concrete state changes, leading to largely deterministic outcomes. We evaluate this capability on LEGO (Lai et al., 2024), where the world model predicts the next visual state based on an image representation of the current state and a textual prompt specifying the action. We follow LEGO to use three contrastive learning based evaluation metrics, CLIP (Radford et al., 2021) score, EgoVLP and EgoVLP+ (Lin et al., 2022), and three common image generation metrics, FID (Heusel et al., 2017), PSNR and LPIPS (Zhang et al., 2018) (with SqueezeNet proposed by Iandola et al., 2016 as the encoder), to make a thorough evaluation.

## 4. Experimental Results

**DynaVieW Excels at Visual Narrative Generation** In Table 2, we present our evaluation results on VinaBench using GPT-4o as the judge.[10] In the zero-shot generalization setting, DynaVieW achieves higher Average score than Emu2 (Sun et al., 2024a), BAGEL (Deng et al., 2025) and a strong baseline Story2Board (Dinkevich et al., 2025) optimized for storyboard creation, driven by clear advantages in cross-scene consistency, while remaining competitive on most per-scene alignment metrics. DynaVieW achieves strong location consistency, which we attribute to extensive pre-training on data that emphasizes visual dynamics at fixed spatial positions.

With access to domain-specific data, DynaVieW (SFT) attains the highest Average score (and also the best or second-best stratified scores) across all SFT baselines, including ARLDM (Pan et al., 2024), MM-Interleaved (Tian et al., 2024), StoryGen (Liu et al., 2024a) and BAGEL. Compared to strong baselines such as BAGEL, DynaVieW shows a more balanced improvement across both per-scene fidelity and long-range consistency, indicating that its schema-guided state-transition modeling effectively structures generation and reduces error accumulation. Compared to zero-

---

[9]We introduce more details of the baseline models, benchmarks and VQA-based evaluation in Appendix B.1, B.2 and B.3.

[10]Results with Gemini-2.5-Pro as the judge are presented in Table 6 in Appendix C.2, which draw the same conclusions.

*Table 2.* Evaluation results of visual narrative generation on VinaBench, with GPT-4o as the judge. "Non-Char. Ent.", "Char. Num." and "Char. Attr." indicate Non-Character Entities, Character Number and Character Attribute, respectively. The best results within each block are **in bold** and the second-best are underlined. More details of different inference settings are included in Appendix B.6.

| Model | Per-Scene Alignment | | | | | Cross-Scene Consistency | | | Average |
|---|---|---|---|---|---|---|---|---|---|
| | Non-Char. Ent. | Char. Num. | Char. Attr. | Time | Location | Style | Character | Location | |
| **Zero-Shot Generalization** | | | | | | | | | |
| Emu2 | 0.533 | 0.220 | 0.556 | 0.523 | 0.466 | 0.378 | 0.352 | 0.439 | 0.425 |
| BAGEL | 0.622 | 0.315 | 0.528 | **0.599** | 0.492 | 0.824 | 0.370 | 0.675 | 0.567 |
| Story2Board | **0.683** | **0.353** | **0.611** | 0.392 | 0.374 | **0.984** | 0.472 | 0.490 | 0.566 |
| DynaVieW | 0.600 | 0.323 | 0.521 | 0.551 | **0.501** | 0.897 | **0.550** | **0.835** | **0.630** |
| **Supervised Fine-Tuning** | | | | | | | | | |
| ARLDM | 0.701 | 0.403 | 0.621 | 0.462 | 0.461 | 0.806 | 0.313 | 0.329 | 0.506 |
| MM-Interleaved | 0.659 | **0.425** | 0.625 | 0.518 | 0.493 | 0.795 | 0.336 | 0.404 | 0.528 |
| StoryGen | 0.566 | 0.388 | 0.412 | 0.313 | 0.416 | 0.576 | 0.158 | 0.341 | 0.389 |
| BAGEL | 0.723 | 0.420 | 0.622 | 0.533 | **0.512** | 0.770 | 0.344 | 0.483 | 0.547 |
| DynaVieW | **0.726** | 0.414 | **0.631** | **0.549** | 0.511 | **0.879** | **0.483** | **0.601** | **0.610** |

*Table 3.* Evaluation results of controllability on visual narrative generation, with GPT-4o as the judge. We use Gemma-3 (27B-it) as an assistant to generate a JSON-schema description of the next scene from multiple aspects, thereby enabling fine-grained control over next-scene image simulation in visual narrative models. Since BAGEL is not pre-trained on JSON data, we further transform each JSON-schema description into natural language (NL). We report the standard deviation across three random seeds. The best results within each block are **in bold** and the second-best results are underlined. More details of generated transitions are included in Appendix B.6.

| Model | Per-Scene Alignment | | | | | Cross-Scene Consistency | | | Average |
|---|---|---|---|---|---|---|---|---|---|
| | Non-Char. Ent. | Char. Num. | Char. Attr. | Time | Location | Style | Char. | Location | |
| BAGEL-NL | 0.355 (± 0.005) | 0.283 (± 0.001) | 0.301 (± 0.003) | 0.433 (± 0.015) | 0.305 (± 0.002) | 0.327 (± 0.010) | 0.108 (± 0.003) | 0.207 (± 0.006) | 0.275 (± 0.001) |
| BAGEL-Schema | 0.355 (± 0.006) | 0.288 (± 0.008) | 0.306 (± 0.007) | 0.485 (± 0.003) | 0.307 (± 0.003) | 0.351 (± 0.012) | 0.109 (± 0.002) | 0.194 (± 0.004) | 0.283 (± 0.002) |
| DynaVieW-NL | 0.530 (± 0.008) | **0.325** (± 0.010) | 0.502 (± 0.004) | 0.493 (± 0.011) | **0.476** (± 0.002) | 0.803 (± 0.012) | **0.588** (± 0.007) | **0.796** (± 0.007) | 0.597 (± 0.002) |
| DynaVieW-Schema | **0.541** (± 0.006) | 0.317 (± 0.004) | **0.509** (± 0.009) | **0.510** (± 0.011) | 0.471 (± 0.004) | **0.858** (± 0.013) | 0.564 (± 0.001) | 0.792 (± 0.013) | **0.604** (± 0.004) |

shot performances, SFT trades off cross-scene consistency scores with higher per-scene alignment scores, indicating richer visual manifestations related to the narrative, instead of coherent but less expressive visual narrative creation. Overall, these results suggest that explicitly modeling the next visual state before generation is particularly beneficial for complex visual narratives that require both fine-grained accuracy and global coherence.

We conduct a head-to-head human evaluation between DynaVieW (SFT) and BAGEL (SFT) to further validate our experiment results. We ask 8 human testers to read the input textual narratives, and then compare storyboards generated by the two models (anonymously presented and randomly ordered). In total, we collected 400 preference votes. The results show that human testers prefer DynaVieW in 47.6% of cases, compared to 35.4% for BAGEL, with 17% of comparisons resulting in a tie. The human evaluation results align well with automatic VLM judges.

**Better Controllability via Hierarchical Visual Dynamic Modeling** To further evaluate the controllability of visual generation, we employ Gemma-3 (27B-it; Team et al., 2025), to predict a JSON-schema description of transition to the next narrative image (or state), exemplified in Figure 2. Compared to the VinaBench narrative constraints generated by SFT models in Table 2, the outputs from Gemma-3 contain more accurate and nuanced changes desired for real-world transition prediction, powered by its better reasoning and instruction-following capability. In Table 3, the evaluation results show that DynaVieW enhanced by LLM-generated JSON schemas substantially outperforms BAGEL across all metrics (0.604 vs 0.283 on average). Note that transforming JSON schemas into natural language (via GPT-4o) does not help with improving BAGEL's overall performance. Its average score remains significantly lower than those of DynaVieW (0.597 vs 0.275), underscoring that the benefit arises from the hierarchical visual dynamic modeling itself rather than the representation format alone. These results demonstrate that hierarchical world modeling provides a more controllable generation process, enabling models to better translate high-level constraints into coherent and faithful visual narratives over time.

**DynaVieW Exhibits Strong Visual Instruction Following Capability** In Table 4, we summarize the instruction following performances of DynaVieW and BAGEL on the LEGO benchmark, which contains two testing subsets

*Table 4.* Results of instruction following in world simulation on LEGO. The best results within each testing subsets are **in bold**.

| | FID ↓ | PSNR ↑ | LPIPS ↓ | CLIP ↑ | EgoVLP ↑ | EgoVLP⁺ ↑ |
|---|---|---|---|---|---|---|
| **Ego4D** | | | | | | |
| BAGEL | 26.94 | 10.64 | 43.57 | 73.78 | 45.07 | 72.61 |
| DynaVieW | 26.15 | 10.74 | 43.47 | 76.09 | 52.88 | 74.54 |
| BAGEL (SFT) | 22.02 | 11.18 | 41.57 | 79.37 | 55.32 | 76.16 |
| DynaVieW (SFT) | **20.96** | **11.43** | **40.77** | **80.99** | **58.73** | **77.62** |
| **Epic-Kitchens** | | | | | | |
| BAGEL | 20.75 | 10.88 | 41.45 | 76.76 | 40.65 | 58.41 |
| DynaVieW | 18.33 | **11.48** | 41.78 | 77.79 | 43.60 | 58.83 |
| BAGEL (SFT) | 11.66 | 11.11 | 39.91 | 82.71 | 47.46 | 61.21 |
| DynaVieW (SFT) | **10.31** | 11.31 | **39.20** | **84.19** | **48.72** | **61.77** |

(Ego4D and Epic-Kitchens). Models are evaluated both before and after SFT on LEGO's task-specific training data. On Ego4D, DynaVieW consistently achieves better performance compared to BAGEL across all evaluation metrics, indicating improvements in both visual fidelity and semantic alignment with instructions. Notably, these gains persist and become more pronounced after SFT. For example, DynaVieW (SFT) attains an FID of 20.96, corresponding to a 19.8% reduction compared to DynaVieW and a 4.8% improvement over BAGEL (SFT), highlighting its ability to follow fine-grained, action-centric instructions in complex egocentric scenarios. The advantage of DynaVieW in world simulation is also evident on Epic-Kitchens, beating BAGEL in nearly all metrics. In particular, DynaVieW surpasses BAGEL on CLIP (77.79 vs 76.76), with the performance gap becoming larger after SFT. Moreover, we validate the superiority of our model on two image-to-text metrics with results reported in Appendix C.3 (Table 7). It indicates that DynaVieW provides a stronger foundation for downstream world simulation tasks. Overall, these results suggest that modeling visual dynamics explicitly enables DynaVieW to better interpret and execute visual instructions, leading to more faithful, semantically aligned, and perceptually coherent outputs. We provide additional comparisons on Epic-Kitchens against baselines from prior work in Appendix C.5. We also present several model output examples in Figure 6 and Figure 7 (Appendix B.4).

**Ablation Study** We conduct a detailed ablation study on the major components of our method: (a) removing multimodal selective attention (w/o *Selective Attention*); (b) removing schema-token re-weighting in the CE loss (w/o *Re-weighted Loss*); (c) replacing the JSON-schema transitions in the training data with natural-language descriptions generated from a heuristic template (w/o *JSON-Schema*); and (d) using a coarser-grained transition schema that omits fine-grained low-level visual transformations and their contributions during training (w/o *Low-level Transitions*). We compare the original DynaVieW with its ablated variants on the downstream VinaBench dataset under the zero-shot generalization setting used in Table 2. As shown in Figure 4, re-

moving multimodal selective attention consistently degrades image-text alignment, while its impact on cross-scene consistency is mixed. Removing schema-token re-weighting in the CE loss also underperforms the full DynaVieW model, albeit with a smaller average drop than removing multimodal selective attention (-0.004 vs. -0.013). Our ablation study on the training data further highlights the importance of both JSON-structured descriptions of visual dynamics and fine-grained hierarchical transitions for effective DynaVieW training, without which leads to average decreases of 0.028 and 0.064, respectively. In particular, using only coarse-grained JSON transitions substantially degrades the model's ability to preserve cross-scene style and character consistency, revealing the central role of fine-grained transition granularity.

# 5. Related Work

**World Modeling** World models are typically tasked with understanding the present state of the human world and predicting its future dynamics (Ding et al., 2025). In particular, the concept of world models is initially proposed with the aim of simulating spatial-temporal representations of game environments, to facilitate reinforcement learning of game agents (Ha & Schmidhuber, 2018). Based on that, the scope of world models is then extended to more real-world environment simulation (Yang et al., 2023), *i.e.*, predicting possible future real-world states as a function of imagined actions (LeCun, 2022). More recent studies have shown that large language models (LLMs) possess promising world modeling potential (Gurnee & Tegmark, 2024), which is closely related to their capabilities of reasoning and planning (Hao et al., 2023), and visual generation (Liu et al., 2024b). LLMs pre-trained with world modeling objectives have been applied to various domains, such as robot learning (Wu et al., 2023), social interaction simulacra (Park et al., 2023) and auto-driving (Gao et al., 2024). However, current world modeling objectives overlook the complex nature of human world visual dynamics, resulting in reliability or controllability deficiencies of multimodal LLMs, and we aim to improve this by integrating a more systematic frame of the visual dynamics into world modeling.

**Visual Dynamics** The study of visual dynamics is crucial for understanding human world activities, from daily life dynamic scenes and events (Pittore et al., 2000; Buxton, 2003) to visual art and narratives (Nienhaus & Dollner, 2005), which also provides an essential ground for image or video frame sequence synthesis and prediction (Xue et al., 2016; Finn et al., 2016). Interpreting the world visual dynamics however requires multiple aspects of visual cognition, including the spatial-temporal grouping of objects (Gepshtein & Kubovy, 2000), the tracking of object state changes (Dickmanns, 2007), the perception of motion and

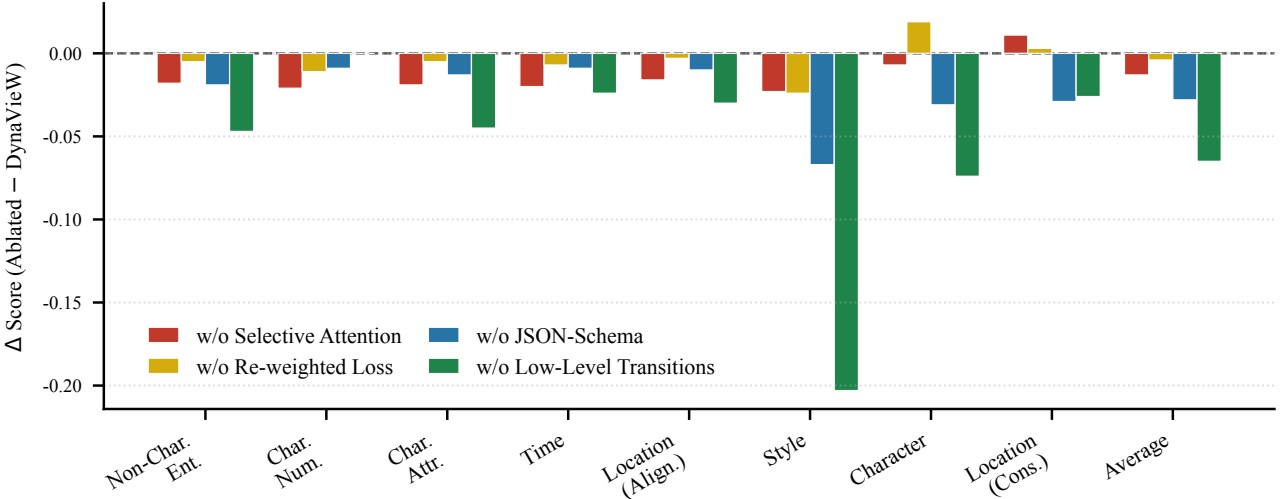

*Figure 4.* **Ablation study of each major component in DynaVieW on VinaBench.** It includes (a) removing the multimodal selective attention (w/o *Selective Attention*), (b) removing the schema token re-weighting of CE loss (w/o *Re-weighted Loss*), (c) using a heuristic template to translate the JSON-schema transitions into natural language descriptions (w/o *JSON-Schema*), and (d) using a more coarse-grained transition schema without describing the fine-grained low-level visual transformations and their contributions (w/o *Low-level Transitions*). Performance change ($\Delta$ score) when removing each component from DynaVieW. Negative values indicate performance drops relative to the full approach.

actions (Johansson, 1975; Bonnet et al., 2005; Dickmanns, 2007), and the modeling of background and camera control (Rowe & Blake, 1995; Yao et al., 2025). We integrate all above aspects of visual dynamic cognition into our hierarchical frame of transition descriptions, to promote more comprehensive understanding of the visual dynamics.

**Interleaved Vision-Text LLMs** Modeling interleaved vision-text data, originally used for multimodal few-shot learning (Alayrac et al., 2022), is currently a typical method of unifying visual understanding and generation (Tian et al., 2024). Early interleaved vision-text LLMs (Tian et al., 2024; Team, 2024; Chern et al., 2024; Sun et al., 2024a; Dong et al., 2024) are pre-trained on public web-sourced image-text sequences (Zhu et al., 2023; Laurençon et al., 2023; Sun et al., 2024b). However, the images in each sequence mostly lack of strong dynamic correlations, *e.g.*, illustrations of unrelated key entities in a paragraph, leading to poor understanding of the visual dynamics required by world modeling. More advanced interleaved vision-text LLMs leverage frames from videos (Soldan et al., 2022; Fan et al., 2022; Bu et al., 2025; Wang et al., 2025) to construct the visual sequences. Nevertheless, their source video types and textual annotations are mostly limited to a single aspect of visual dynamic cognition, such as high-level progression of movie plots (Lin et al., 2024a), viewpoint changes in game environments (Zhang et al., 2025; He et al., 2025), actions of robots (Qu et al., 2025) and coarse-grained captions of scene shifts (Deng et al., 2025). In this work, we build broader and more comprehensively annotated video dynamic data to elevate the world modeling capability of interleaved vision-text LLMs.

## 6. Conclusion

In this paper, we propose a new world model, DynaVieW, to address the challenge of complex visual dynamic understanding and generation. DynaVieW builds in-depth comprehension of dynamic visual scenes in human world activities via a joint learning of video state simulation and hierarchical state transition prediction. In downstream experiments, DynaVieW demonstrates superior capabilities of controllable visual creation and instruction following. Our results highlight the positive impact of improved world modeling on the downstream task performances of multimodal LLMs. This motivates a promising future direction of developing more powerful foundation world models, which can serve as reliable backbones of multimodal applications.

## Acknowledgements

We gratefully acknowledge the support of the Swiss National Science Foundation (No. 215390), the European Research Council (Starting grant no. 101222478, RESPECT-LM), the AI2050 program at Schmidt Sciences (Grant #G-25-69783), Sony Group Corporation, and the Swiss National Supercomputing Center (CSCS) in the form of an infrastructure engineering and development project. This research is with support from Google.org and the Google Cloud Research Credits program for the Gemini Academic

Program. This work was supported as part of the Swiss AI Initiative by a grant from the Swiss National Supercomputing Centre (CSCS) under project ID a173 on Alps. Li Mi gratefully acknowledges the support of the Horizon Europe grant 101213369 DVPS.

## Impact Statement

This paper presents work whose goal is to advance the field of Multimodal Large Language Models in Artificial Intelligence. There are many potential societal consequences of our work, none of which we feel must be specifically highlighted here.

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

# A. DynaVieW Implementation Details

## A.1. State-Transition Data Construction Details

**State Extraction**    We extract our world modeling states as keyframes from three source video datasets, including Ego4D (Grauman et al., 2022), AgiBotWorld-Alpha (Bu et al., 2025) and five sub-portions (Panda-70M, Pexels, Pixabay, Mixkit, BDD100K) of ShareGPT4Video (Chen et al., 2024a). We control the number of videos selected from each source dataset based on a balanced number of extracted states ($\sim 53K$ states from each source). Note that videos from Ego4D are mostly very long, which can reach to 30 minutes, resulting in a large number of states extracted from just a single Ego4D video. To include more diverse Ego4D videos, while maintain their number of extracted states to be comparable with the other two sources, we sample a sub-segment of each Ego4D video (instead of the whole video) to perform the state extraction, according to the video segmentation annotated by Ego4D-HCap (Islam et al., 2024). Table 5 includes the detailed descriptions of each source dataset (or sub-portion) and the statistics of our state extraction.

**Transition Annotation**    We prompt an oracle vision language model (VLM), InternVL-78B-Instruct (Chen et al., 2024b), to annotate the transition between each pair of adjacent states (or keyframes) extracted from our source videos. InternVL is first instructed to generate a dense caption of the former state, to understand the visual contents presented at the start of the transition. Based on that, InternVL is then prompted by a fine-grained task demonstration to comprehensively analyze the transition following our defined schema in Sec 2.1. The prompts for dense caption and transition analysis are below:

---

**Oracle VLM Prompt for Dense Caption**

You are given an image. Your goal is to provide a concise but detailed caption for the image. Your caption should address all the objects in the scene and their spatial relationships to each other.

Follow these quality standards:

Precision: Use specific, measurable descriptions rather than vague qualifiers.
Accuracy: Ensure all observations are factually correct and verifiable through direct visual inspection. Do not use suggestive language about the image (e.g."suggesting a point-of-view angle..." or "indicating that..."). Do not describe anything beyond what is immediately provided in the image.
Evidence-Based Grounding: Base every claim on observable pixel-level evidence in the frames - cite specific visual details, colors, shapes, positions, and changes that support your analysis.
Completeness: Address all relevant aspects of the scene.
Consistency: Maintain consistent terminology and granularity throughout the analysis.
Relevance: Focus on description that meaningfully contributes to the task while being concise. If you are unsure of what a specific object in the scene is, avoid referring to it.
Clarity: Ensure descriptions are unambiguous and interpretable by other systems.

Final Output Format:
Enclose your caption within a structured JSON output following this exact schema:
```
{
    "type": "object",
    "properties": {
        "caption": {"type": "string"}
    }
}
```

Remember: Your goal is to provide a concise, yet detailed caption for the image you are given. Your caption must be specific and factually correct, containing no erroneous statements about the image.
Do not describe anything beyond what is immediately provided in the image. Everything must be factually correct and verifiable from the image. Be concise. If you are unsure about something, simply do not say it.

Here is the image: ⟨the former state image⟩

---

*Table 5.* Detailed statistics of state extraction in DynaVieW training data, including number of videos (**# Videos**) selected from each source dataset (or sub-portion), average (**Avg.**), minimum (**Min**) and maximum (**Max**) **Duration** of a video in seconds, total number of states (**# States**) extracted from each source dataset, average, minimum and maximum number of states extracted from per video (**States per Video**), average, minimum and maximum time interval in seconds between each pair of adjacent states (**State Interval**).

| Source | Portion | Domain | # Videos | Duration (s) | | | # States | States per Video | | | State Interval (s) | | |
|---|---|---|---|---|---|---|---|---|---|---|---|---|---|
| | | | | Avg. | Min | Max | | Avg. | Min | Max | Avg. | Min | Max |
| Ego4D(-HCap) | - | ego-centric human daily life (household, outdoor, workplace, leisure, etc.) activities | 1000 | 117.06 | 2.10 | 309.90 | 52976 | 52.98 | 2 | 100 | 2.25 | 0.80 | 167.10 |
| AgiBotWorld-Alpha | - | robot-view trajectories in domestic, retail, industrial, restaurant, and office environment | 3400 | 46.09 | 1.07 | 80.07 | 53282 | 15.67 | 2 | 60 | 3.14 | 0.80 | 74.10 |
| ShareGPT4Video | Panda-70M | public YouTube videos (gaming, TV shows, *etc.*) | 1200 | 23.97 | 0.90 | 49.15 | 13215 | 11.01 | 2 | 39 | 2.39 | 0.80 | 45.05 |
| | Pexels Pixabay Mixkit | aesthetically appealing stock videos (natural landscape, cultural scenery,*etc.*) | 1800 1800 572 | 19.12 20.28 13.84 | 0.83 0.83 0.96 | 274.08 109.0 67.07 | 12871 11999 2248 | 7.15 6.67 3.93 | 2 2 2 | 38 40 24 | 3.11 3.58 4.72 | 0.80 0.80 0.84 | 56.06 58.10 32.03 |
| | BDD100K | auto-driving video recordings | 603 | 36.32 | 3.02 | 41.93 | 13015 | 21.58 | 2 | 40 | 1.76 | 0.80 | 39.87 |
| | All | - | 5975 | 21.67 | 0.83 | 274.08 | 53348 | 8.93 | 2 | 40 | 2.73 | 0.80 | 58.10 |

---

**Oracle VLM Prompt for Transition Analysis (Part 1)**

Your core objective is to generate comprehensive descriptions of a visual transition that ACCURATELY captures the dynamic transformation between two consecutive video frames, focusing on fine-grained visual features and their temporal relationships. You should precisely capture what happens in the transition between two frames, only reporting verifiable descriptions you are entirely sure about and the frames give you clear evidence for. Do not extrapolate hypothetical possibilities based on what you infer from the frames, only report exactly what you see. Ensure that all observations you make are factually correct and verifiable through direct visual inspection.

Analysis Framework

Step 1: High-Level Activity Identification

First, identify the overarching activity or scenario occurring between the two frames. This should be a broad categorization that encompasses the primary action or event taking place. Examples: "Person preparing a meal in kitchen", "Vehicle navigating through traffic", "Athletes competing in a sports event", "Construction work in progress".

Step 2: Sub-Activity Decomposition

Break down the high-level activity into constituent sub-activities. These are intermediate-level actions that contribute to the overall activity. Only list sub-activities that are visibly occurring in the given frames. Do not list any sub-activities that you do not have clear evidence for. You must NOT invent any sub-activities that are not immediately verifiable from the frames – for example, if there is a knife and a carrot in the scene, unless the image clearly shows that the knife is being used to chop the carrot, you cannot say anything about the knife potentially being used to chop the carrot. Examples for "Person preparing a meal": "Gathering ingredients from pantry", "Chopping vegetables on cutting board", "Heating pan on stove", "Combining ingredients in bowl".

Step 3: Atomic Action Identification

For each sub-activity, identify the atomic actions - the smallest meaningful units of action that cannot be further decomposed while maintaining semantic meaning. Only list atomic actions that actually occur in between the frames! For example, if a bell pepper is not actually lifted or moved, DO NOT say anything about the bell pepper. Examples for "Chopping vegetables on cutting board": "Positioning knife above carrot", "Applying downward pressure to slice", "Lifting knife blade", "Sliding carrot segment aside".

Step 4: Transformation Analysis

For each atomic action, systematically analyze the following six categories of transformations:

4.1 Object Transformations

New Objects Introduced: Identify objects that appear in Frame 2 but were not present or visible in Frame 1.
Objects Removed: Identify objects that disappear, become occluded, or move out of frame.
Object Persistence: Note objects that remain present but may undergo other changes.

**Oracle VLM Prompt for Transition Analysis (Part 2)**

4.2 Object State Transformations

Color Changes: Variations in hue, saturation, brightness, or lighting conditions affecting object appearance.
Shape Deformation: Changes in object geometry, size, orientation, or physical configuration.
Physical Status: Alterations in object condition (broken/intact, open/closed, full/empty, wet/dry, *etc.*).
Texture Changes: Modifications in surface appearance or material properties.

4.3 Spatial Relation Transformations

Positional Relationships: Changes in relative positioning between objects (above/below, left/right, front/back).
Contact Relationships: Modifications in physical contact or proximity (touching/separated, inside/outside, attached/detached).
Alignment Changes: Shifts in object alignment, orientation, or arrangement patterns.

4.4 Action Transformations

Action Continuation: Ongoing actions that persist from Frame 1 to Frame 2.
Action Completion: Actions that conclude or reach a terminal state.
Action Initiation: New actions that begin in Frame 2.
Action Interruption: Actions that pause, stop, or are disrupted.

4.5 Motion Transformations

Translational Motion: Linear movement in any direction (forward, backward, sideways, up, down).
Rotational Motion: Spinning, turning, or rotating movement around an axis.
Oscillatory Motion: Back-and-forth or periodic movement patterns.
Deformation Motion: Changes in object shape through stretching, compression, or bending.

4.6 Camera Transformations

Viewpoint Changes: Modifications in camera position or angle.
Zoom Operations: Magnification changes (zoom in/out).
Pan/Tilt Movements: Horizontal or vertical camera sweeping.
Focus Adjustments: Changes in depth of field or focal point.

4.7 Background Transformations

Illumination Changes: Variations in lighting conditions, shadows, or brightness.
Scene Shifts: Changes in background environment or setting.
Atmospheric Conditions: Modifications in weather, visibility, or environmental factors.
Contextual Elements: Changes in background objects or environmental details.

Step 5: Contribution Analysis

For the identified transformations of each atomic action, analyze and explain:
Atomic Action Contribution: How the transformations directly enable, facilitate, or result from the specific atomic action.
Sub-Activity Contribution: How the transformations support or advance the broader sub-activity goal.
Temporal Significance: The timing and sequence importance of the transformations within the overall activity flow.

Quality Standards:

Accuracy: Ensure all observations are factually correct and verifiable through direct visual inspection.
Evidence-Based Grounding: Base every claim on observable pixel-level evidence in the frames - cite specific visual details, colors, shapes, positions, and changes that support your analysis.
Precision: Use specific, measurable descriptions rather than vague qualifiers.
Completeness: Address all relevant transformation categories.
Consistency: Maintain consistent terminology and granularity throughout the analysis.
Relevance: Focus on transformations that meaningfully contribute to the activity progression.
Clarity: Ensure descriptions are unambiguous and interpretable by other systems.

**Oracle VLM Prompt for Transition Analysis (Part 3)**

Output Requirements

Reasoning Process – Think through your analysis step-by-step, only considering activities and actions you have clear evidence for, and documenting your reasoning for each level of the framework. Show your work in identifying activities, sub-activities, atomic actions, and transitions in a JSON as described below. Do not output anything other than the final JSON as described.

Final Output Format – Conclude your analysis with a structured JSON output following this exact schema:

```
{"type": "object", "properties": {
"high_level_activity": {"type": "string"},
   "sub_activities": {"type": "array", "items": {"type": "object", "properties": {"name": {"type": "string"},
     "atomic_actions": {"type": "array", "items": {"type": "object", "properties": {"name": {"type": "string"},
       "transformations": {"type": "object", "properties": {
         "objects": {"type": "object", "properties": {
           "introduced": {"type": "array", "items": {"type": "string"}},
           "removed": {"type": "array", "items": {"type": "string"}},
           "persistent": {"type": "array", "items": {"type": "string"}}}},
         "object_states": {"type": "object", "properties": {
           "color_changes": {"type": "array", "items": {"type": "string"}},
           "shape_changes": {"type": "array", "items": {"type": "string"}},
           "physical_status_changes": {"type": "array", "items": {"type": "string"}},
           "texture_changes": {"type": "array", "items": {"type": "string"}}}},
         "spatial_relations": {"type": "object", "properties": {
           "positional_changes": {"type": "array", "items": {"type": "string"}},
           "contact_changes": {"type": "array", "items": {"type": "string"}},
           "alignment_changes": {"type": "array","items": {"type": "string"}}}},
         "actions": {"type": "object","properties": {
           "continuing": {"type": "array", "items": {"type": "string"}},
           "completing": {"type": "array", "items": {"type": "string"}},
           "initiating": {"type": "array", "items": {"type": "string"}},
           "interrupting": {"type": "array", "items": {"type": "string"}}}},
         "motion": {"type": "object", "properties": {
           "translational": {"type": "array", "items": {"type": "string"}},
           "rotational": {"type": "array", "items": {"type": "string"}},
           "oscillatory": {"type": "array", "items": {"type": "string"}},
           "deformation": {"type": "array", "items": {"type": "string"}}}},
         "camera": {"type": "object", "properties": {
           "viewpoint_changes": {"type": "array", "items": {"type": "string"}},
           "zoom_changes": {"type": "array", "items": {"type": "string"}},
           "pan_tilt": {"type": "array", "items": {"type": "string"}},
           "focus_changes": {"type": "array", "items": {"type": "string"}}}},
         "background": {"type": "object", "properties": {
           "illumination_changes": {"type": "array", "items": {"type": "string"}},
           "scene_shifts": {"type": "array", "items": {"type": "string"}},
           "atmospheric_changes": {"type": "array", "items": {"type": "string"}},
           "contextual_changes": {"type": "array", "items": {"type": "string"}}}}}},
       "contributions": {"type": "object", "properties": {
         "to_atomic_action": {"type": "string"},
         "to_sub_activity": {"type": "string"},
         "temporal_significance": {"type": "string"}}}
     }}}
   }}}
}}
```

---

**Oracle VLM Prompt for Transition Analysis (Part 4)**

Error Handling

If certain transition categories are not applicable, do not include them in the JSON at all.

If the relationship between frames is unclear, state your assumptions explicitly. Try to avoid making any assumptions that are not factually correct and verifiable – it is better to have less information that is more reliable than vice versa. If objects are partially occluded, specify the degree of visibility and confidence level.

Remember: Your goal is to create a comprehensive, structured understanding of visual change that captures both the obvious and subtle transformations occurring between the two frames. All your observations must be factually correct and verifiable through direct visual inspection – you must strive for reliable, accurate information ONLY. You must base every claim on observable pixel-level evidence in the frames.

Here is the first frame: ⟨the former state image⟩
To guide you, here is a caption of the first frame: ⟨dense caption of the former state image⟩
Here is the second frame: ⟨the latter state image⟩

---

We adopt constrained decoding (Hokamp & Liu, 2017) within vLLM (Kwon et al., 2023) infrastructure to ensure that the transition generated by the oracle VLM is in valid JSON structure and follows our defined schema in above transition analysis prompt (Part 3). Figure 5 presents an example of our state transition annotation based on our defined JSON schema.

### A.2. DynaVieW Modeling Details

We initialize DynaVieW with the pre-trained weights of BAGEL-7B-MoT (Deng et al., 2025) and conduct continued pre-training on our state-transition data.

**Data Preprocessing** To fit our maximum computing (GPU) memory, we split each long state-transition sequence into sub-sequences with a maximum length of 6 states interleaved with 5 transitions in between, where adjacent sub-sequences have an overlap of 3 states, *i.e.*, the sub-sequence sliding window has a width of 6 states and a stride of 3 states. After the split, we get a total number of 48260 state-transition (sub-)sequences as our final training samples. To mitigate catastrophic forgetting, we further mix our state-transition data with 3000 BAGEL's original pre-training data samples[11], including text-to-image generation, multi-round image-editing and LLaVA-OneVision (Li et al., 2024) modeling samples (1000 each). For each state image encoded by the VAE encoder (FLUX.1-dev[12], with patch size 16), which is used for the diffusion training of state simulation, we resize the image to ensure that its longer side (height or width) is in the range of 512 to 1024 pixels, while keeping its original aspect ratio. For each state image encoded by the ViT encoder (SigLIP2-so400m-patch14-384[13], with patch size 14), which serves as contexts for predicting subsequent transitions and states, we resize the image to ensure that its longer side fall in the range of 224 to 518 pixels, also under the original aspect ratio.

**Training Hyperparameters and Computing Resources** We train DynaVieW on a cosine learning rate scheduler, with warm-up steps, total training steps, maximum learning rate and minimum learning rate set to 1000, 10000, $5e^{-5}$ and $1e^{-6}$, respectively. We deploy the training on 32 NVIDIA GH200 GPUs, which takes about 4 days. To improve training efficiency, we follow BAGEL to use a dynamic batch size of training samples yielded to each GPU at each training step. Specifically, we keep adding data samples into a batch until its total number of tokens reaches 16384 or above, before yielding the batch to a GPU. However, to avoid out of GPU memory, if the total number of tokens jumps directly from less than 16384 to more than 36864 (the maximum number of tokens allowed in a batch to fit our GPU memory) after adding a sample, we drop this sample to the next batch and iterate to add the next sample into the current batch. Same as the BAGEL setting, we freeze the weights of the VAE encoder during the training (and the VAE decoder which is only used at the inference phase), while update the rest parts of the model. The AdamW (Loshchilov & Hutter, 2018) optimizer is used, with $\beta_1$, $\beta_2$ and $\epsilon$ set to 0.9, 0.95 and $1e^{-15}$, respectively, and the maximum norm for gradient clipping set to 1.0.

---

[11] https://github.com/ByteDance-Seed/Bagel/blob/main/TRAIN.md
[12] https://huggingface.co/black-forest-labs/FLUX.1-dev
[13] https://huggingface.co/google/siglip2-so400m-patch14-384

```
{"high_level_activity": "Sitting in a living room, relaxing and playing the guitar",
  ↪"sub_activities": [{"name": "Playing the guitar",
    ↪"atomic_actions": [{"name": "Positioning guitar",
      ↪"transformations": {
        ↪"objects": {"introduced": ["guitar", "person with guitar"], "removed": [], "persistent": ["laptop", "coffee table", "potted plant", "person on sofa"]},
        ↪"object_states": {"color_changes": [], "shape_changes": [], "physical_status_changes": [], "texture_changes": []},
        ↪"spatial_relations": {"positional_changes": ["Guitar situating next to a laptop"], "contact_changes": ["A person holding a guitar"], "alignment_changes": []},
        ↪"actions": {"continuing": [], "completing": [], "initiating": ["Playing the guitar"], "interrupting": []},
        ↪"motion": {"translational": [], "rotational": [], "oscillatory": [], "deformation": []},
        ↪"camera": {"viewpoint_changes": [], "zoom_changes": [], "pan_tilt": ["The camera moves to show the person playing the guitar."], "focus_changes": []},
        ↪"background": {"illumination_changes": [], "scene_shifts": [], "atmospheric_changes": [], "contextual_changes": []}},
      ↪"contributions": {
        ↪"to_atomic_action": "Showing the person with the guitar explains why there is a tutorial video.",
        ↪"to_sub_activity": "The situating of the guitar starts the sub-activity of playing the guitar.",
        ↪"temporal_significance": "The transition shows the start of the guitar playing that can be seen in the video."}}, ...]}, ...]}
```

*Figure 5.* An example of DynaVieW state transition. We frame our transition based on a hierarchical JSON schema.

## A.3. DynaVieW Validation Details

We use GPT-4o (Hurst et al., 2024; Achiam et al., 2023) as a VLM judge to validate DynaVieW's performances of transition prediction and state simulation, and meanwhile to verify the quality of gold transitions and states constructed by our pipeline, based on the 900 validation samples (300 each from Ego4D, AgiBotWorld-Alpha and ShareGPT4Video datasets). The VLM judge prompts used for the transition and state validation are shown below:

---

**VLM Judge Prompt for Transition Validation (Part 1)**

You are given a sequence of keyframes of a video. Your core objective is to evaluate the reasonability of a response that predicts the dynamic transformation from the last given keyframe to the next unknown keyframe. The response are in JSON format. When conducting your evaluation of the response, you must focus on the following aspects:

1. High-Level Activity: Whether the "high_level_activity" field reasonably predicts what will happen or continue to occur from the last given keyframe to the next keyframe? Answer "yes" or "no". Also output a concise justification of your answer in one or two sentences.

2. Sub-Activities: Whether all the "sub_activities" listed in the response reasonably predict what will happen or continue to occur from the last given keyframe to the next keyframe? Answer "yes" or "no" according to the "name" of each sub-activity. Also output a concise justification of your answer in one or two sentences.

3. Atomic Actions: Whether all the "atomic_actions" of sub-activities reasonably predict what will happen or continue to occur from the last given keyframe to the next keyframe? Answer "yes" or "no" according to the "name" of each atomic action. Also output a concise justification of your answer in one or two sentences.

4.1. Transformations – Objects: Whether the "objects" field of the "transformations" reasonably predicts the objects that will be newly "introduced", "removed" or kept "persistent" from the last given keyframe to the next keyframe? Answer "empty" if all lists in this field are empty. Otherwise, answer "yes" or "no" according to the predictions in non-empty lists. Also output a concise justification of your answer in one or two sentences.

4.2. Transformations – Object States: Whether the "object_states" field of the "transformations" reasonably predicts the changes of object "color", "shape", "physical status" or "texture" from the last given keyframe to the next keyframe? Answer "empty" if all lists in this field are empty. Otherwise, answer "yes" or "no" according to the predictions in non-empty lists. Also output a concise justification of your answer in one or two sentences.

---

**VLM Judge Prompt for Transition Validation (Part 2)**

4.3. Transformations – Spatial Relations: Whether the "spatial_relations" field of the "transformations" reasonably predicts the "positional", "contact" or "alignment" changes of objects from the last given keyframe to the next keyframe? Answer "empty" if all lists in this field are empty. Otherwise, answer "yes" or "no" according to the predictions in non-empty lists. Also output a concise justification of your answer in one or two sentences.

4.4. Transformations – Actions: Whether the "actions" field of the "transformations" reasonably predicts the "continuing", "completing", "initiating" or "interrupting" actions of objects from the last given keyframe to the next keyframe? Answer "empty" if all lists in this field are empty. Otherwise, answer "yes" or "no" according to the predictions in non-empty lists. Also output a concise justification of your answer in one or two sentences.

4.5. Transformations – Motion: Whether the "motion" field of the "transformations" reasonably predicts the "translational", "rotational", "oscillatory" or "deformation" motion of objects from the last given keyframe to the next keyframe? Answer "empty" if all lists in this field are empty. Otherwise, answer "yes" or "no" according to the predictions in non-empty lists. Also output a concise justification of your answer in one or two sentences.

4.6. Transformations - Camera: Whether the "camera" field of the "transformations" reasonably predicts the "viewpoint", "zoom", "focus" changes or "pan tilt" of the camera from the last given keyframe to the next keyframe? Answer "empty" if all lists in this field are empty. Otherwise, answer "yes" or "no" according to the predictions in non-empty lists. Also output a concise justification of your answer in one or two sentences.

4.7. Transformations - Background: Whether the "background" field of the "transformations" reasonably predicts the "illumination", "atmospheric", "contextual" changes or "scene_shifts" of the background from the last given keyframe to the next keyframe? Answer "empty" if all lists in this field are empty. Otherwise, answer "yes" or "no" according to the predictions in non-empty lists. Also output a concise justification of your answer in one or two sentences.

5. Contributions: Whether the "contributions" field reasonably analyzes the contributions of predicted "transformations" to "atomic_actions" and "sub_activities", and the "temporal_significance" of predicted "transformations"? Answer "yes" or "no". Also output a concise justification of your answer in one or two sentences.

Final Output Format

Conclude your evaluation with a structured JSON output following this exact schema:

{
"high_level_activity":{"reasonable": "yes" or "no", "justification": "string"},
"sub_activities":{"reasonable": "yes" or "no", "justification": "string"},
"atomic_actions":{"reasonable": "yes" or "no", "justification": "string"},
"transformation_objects":{"reasonable": "yes", "no" or "empty", "justification": "string"},
"transformation_object_states":{"reasonable": "yes", "no" or "empty", "justification": "string"},
"transformation_spatial_relations":{"reasonable": "yes", "no" or "empty", "justification": "string"},
"transformation_actions":{"reasonable": "yes", "no" or "empty", "justification": "string"},
"transformation_motion":{"reasonable": "yes", "no" or "empty", "justification": "string"},
"transformation_camera":{"reasonable": "yes", "no" or "empty", "justification": "string"},
"transformation_background":{"reasonable": "yes", "no" or "empty", "justification": "string"},
"contributions":{"reasonable": "yes" or "no", "justification": "string"}
}

Do not output anything other than the final JSON as described.

Here is the sequence of video keyframes: ⟨preceding state images before the transition to be evaluated⟩

Here is the response to evaluate: ⟨the transition to be evaluated⟩

**VLM Judge Prompt for State Validation**

You are given a sequence of keyframes of a video. Your core objective is to evaluate a prediction of the next keyframe in the video. When conducting your evaluation of the next keyframe prediction, you must focus on the following aspects:

1. Style Consistency: Whether the image style of the next keyframe is consistent with former keyframes? Answer an integer from 1 to 10, to indicate the consistency level, where 1 means totally inconsistent image style and 10 means totally the same image style. Also output a concise justification of your answer in one or two sentences.

2.1. Transformations - Objects: From the former keyframe to the next keyframe, whether all the appearance, disappearance or retention of objects accord with spatial-temporal causality and commonsense? Answer "yes" or "no". Also output a concise justification of your answer in one or two sentences.

2.2. Transformations - Object States: From the former keyframe to the next keyframe, whether all the changes of objects' color, shape, texture or physical status accord with spatial-temporal causality and commonsense? Ignore the changes caused mainly by image style shifts. If there is no change of object states, answer "unchange". Otherwise, answer "yes" or "no" according to the changes. Also output a concise justification of your answer in one or two sentences.

2.3. Transformations - Spatial Relations: From the former keyframe to the next keyframe, whether all the changes of objects' alignment, contact or positional relations accord with spatial-temporal causality and commonsense? If there is no change of spatial relations, answer "unchange". Otherwise, answer "yes" or "no" according to the changes. Also output a concise justification of your answer in one or two sentences.

2.4. Transformations - Motion or Actions: From the former keyframe to the next keyframe, whether all the changes or progressions of object motion or actions accord with spatial-temporal causality and commonsense? If there is no change or progression of motion or actions, answer "unchange". Otherwise, answer "yes" or "no" according to the changes. Also output a concise justification of your answer in one or two sentences.

2.5. Transformations - Camera: From the former keyframe to the next keyframe, whether all the changes of camera viewpoint, zoom, focus or pan tilt accord with spatial-temporal causality and commonsense? If there is no change of camera, answer "unchange". Otherwise, answer "yes" or "no" according to the changes. Also output a concise justification of your answer in one or two sentences.

2.6. Transformations - Background: From the former keyframe to the next keyframe, whether all the changes of background illumination, atmosphere, environment or setting accord with spatial-temporal causality and commonsense? Ignore the changes caused mainly by image style shifts. If there is no change of background, answer "unchange". Otherwise, answer "yes" or "no" according to the changes. Also output a concise justification of your answer in one or two sentences.

Final Output Format

Conclude your evaluation with a structured JSON output following this exact schema:

{
"style_consistency":{"rating": integer, "justification": "string"},
"transformation_objects":{"accordance": "yes" or "no", "justification": "string"},
"transformation_object_states":{"accordance": "yes", "no" or "unchange", "justification": "string"},
"transformation_spatial_relations":{"accordance": "yes", "no" or "unchange", "justification": "string"},
"transformation_motion_actions":{"accordance": "yes", "no" or "unchange", "justification": "string"},
"transformation_camera":{"accordance": "yes", "no" or "unchange", "justification": "string"},
"transformation_background":{"accordance": "yes", "no" or "unchange", "justification": "string"}
}
Do not output anything other than the final JSON as described.

Here is the sequence of former video keyframes: ⟨preceding state images before the state to be evaluated⟩

Here is the next keyframe prediction to evaluate: ⟨the state image to be evaluated⟩

# B. Experimental Details

## B.1. Baseline Methods

**Emu2** (Sun et al., 2024a) is a 37B-parameter generative multimodal foundation model designed to support both multimodal understanding and generation through in-context learning. It is trained on large-scale multimodal sequences, including text, image-text pairs, video-text pairs, and interleaved image-text-video data, using a unified autoregressive objective that predicts the next multimodal element, either a text token or a visual embedding. Architecturally, Emu2 consists of a visual encoder, a Transformer-based multimodal modeling module, and a visual decoder: images are encoded into continuous visual embeddings, interleaved with text tokens, modeled autoregressively, and then decoded back into images or videos when generation is required.

**BAGEL** is an open-sourced multimodal foundation model with 7B active parameters (14B total) trained on large-scale interleaved multimodal data, including text, image, video, and web data. It adopts a Mixture-of-Transformer-Experts (MoT) architecture that employs selective activation of modality-specific parameters and unifies multimodal understanding (i.e., understanding expert) and generation (i.e., generation expert) through shared self-attention operations. For visual understanding, it leverages a ViT (Dosovitskiy, 2020) encoder, initialized by SigLIP2-so400m/14 (Tschannen et al., 2025) with a fixed 384-resolution, to convert the raw pixels into tokens. A two-layer MLP connector is adopted to match the feature dimension of the ViT tokens and the LLM hidden states. For visual generation, it uses a pre-trained VAE model from FLUX (Labs, 2024) to convert images from pixel space to latent space. The VAE encoder is frozen during training.

**Story2Board** (Dinkevich et al., 2025) is a training-free framework for expressive storyboard creation from natural language, with a special focus on enhancing visual consistency. It chooses FLUX.1-dev (Labs, 2024), a dense rectified flow transformer with 12B parameters, as the underlying VLM for storyboard generation. On top of FLUX.1-dev, it proposes two consistency-enhancing techniques: Latent Panel Anchoring, which preserves a shared character reference across panels, and Reciprocal Attention Value Mixing, which softly blends visual features between token pairs with strong reciprocal attention. For a fair comparison with other baselines, we always use the first ground-truth image of the storyboard as the reference panel at the top and the first narrative as the reference panel prompt, while the other transition narratives are panel prompts for different scenes.

**ARLDM** (Pan et al., 2024) trains a Stable Diffusion (Rombach et al., 2022) module to auto-regressively generate each visual narrative image, which is conditioned on the BLIP (Li et al., 2022) embeddings of previous scenes' generated images and input textual constraints, and the CLIP (Radford et al., 2021) embedding of the next scene's input textual constraints.

**MM-Interleaved** (Tian et al., 2024) trains a VLM, *i.e.*, Vicuna (Zheng et al., 2023) with CLIP vision encoder, to model the interleaved sequence of previously generated images and their textual constraints, and a Stable Diffusion module to generate the next narrative image based on the output states of the VLM. Both the VLM and the diffusion module are augmented by additional layers of cross-attention to sparse image features via Deformable Attention (Zhu et al., 2020).

**StoryGen** (Liu et al., 2024a) uses a dual-diffusion structure to perform the auto-regressive generation of narrative images. It first adds noise to each previously generated image, and then the noisy image is de-noised by a Stable Diffusion module (conditioned on the image's corresponding input textual constraints), whose latent diffusion states are used as the extracted features of the image. Conditioned on the next textual constraints and the concatenation of previous images' extracted features, a second Stable Diffusion module is trained to generate the next narrative image.

## B.2. Evaluation Benchmarks

**VinaBench** VinaBench (Gao et al., 2025) is a novel benchmark proposed for visual narrative generation, consisting of 25K pairs of visual and textual narratives sampled from diverse visual storytelling datasets. VinaBench contains rich annotations about commonsense links between textual and visual narratives. Moreover, VinaBench provides a set of global and scene-specific feature annotations to explicitly reveal the visual discourse. In this work, we primarily focus on the Visual Writing Prompts (VWP) subset that has 834 high-quality storyboards for downstream evaluations on visual narrative generation. We convert part of the annotations into the form of JSON schema, which is then used in supervised fine-tuning as part of textual transition descriptions. VinaBench proposes 5 metrics to measure the fine-grained alignment of each generated image with its corresponding scene's annotated narrative constraints: Non-Character Entities, Character Number,

Character Attribute, Time of Day, and Location. For each narrative sample, VinaBench adopts 3 metrics to assess the consistency of generated visual narrative images in terms of Style, Character, and Location. The prompt details can be found in Appendix B.3. We present the VinaBench evaluation results with GPT-4o as the judge in Table 2, and further validate the evaluation results by using Gemini-2.5-Pro as the judge in Table 6.

**LEGO** LEGO (Lai et al., 2024) introduces the egocentric action frame generation task, synthesizing an image depicting an action in the user's context (i.e., action frame) by conditioning on a user prompt and an input egocentric image. LEGO contains two egocentric action datasets, Ego4D and Epic-Kitchens, both of which are densely annotated with action starting time $t$ and ending time $\hat{t}$. The LEGO dataset is constructed by selecting an egocentric image frame $\delta_i$ seconds before the action begins as the input $\mathcal{X}$, and an image $\delta_o$ seconds after the action begins as the target frame $\mathcal{Y}$. After data filtering, LEGO ultimately contains 85,521/9,931 data samples for the train/test sets from Ego4D and 61,841/8,893 data samples for the train/test sets from Epic-Kitchens. LEGO evaluates the action instruction-following performance in real-world simulation via 6 image-to-image similarity metrics. It includes (1) EgoVLP score (Lin et al., 2022), (2) EgoVLP$^+$ score, and (3) CLIP score (Radford et al., 2021), all of which are based on contrastive learning. They also report (4) Fréchet Inception Distance (FID) (Heusel et al., 2017), (5) Peak Signal-to-Noise Ratio (PSNR), and (6) Learned Perceptual Image Patch Similarity (LPIPS) (Zhang et al., 2018). We report the performance of our model and baselines using these metrics in Table 4. In addition, we adopt two image-to-text metrics, BLIP-B(ase) and BLIP-L(arge), which utilize BLIP models to generate image captions of the output images and then calculate text-to-text similarity using the CLIP text encoder. We summarize results under image-to-text metrics in Table 7.

### B.3. VLM Judge Prompts on VinaBench

In this section, we present the complete set of prompts used by the VLM judges on VinaBench, covering both per-scene alignment evaluation and cross-scene consistency evaluation.

**Per-Scene Alignment.**

- **Character Number.** At each timestamp, we provide the generated image together with the ground-truth character count from VinaBench annotations and ask the judge to identify the number of characters appearing in the image. The judge is required to respond using Arabic numerals only.

- **Character Attributes.** To assess fine-grained character fidelity, we include character–description pairs in the prompt and ask the judge whether the characters in the generated image match their corresponding descriptions. If all characters are consistent with the annotations, the judge answers "Yes"; otherwise, it answers "No".

- **Non-character Entities.** Beyond characters, we evaluate non-character entities by asking whether the generated image contains or implies a specific target entity. The judge responds "Yes" or "No" for each entity, and we compute the proportion of positive responses.

- **Location.** We directly ask the judge whether the image is taken at a particular location specified by the ground-truth annotation. The judge responds "Yes" or "No".

- **Time.** The judge is prompted to determine whether the image corresponds to a specific time or time period described in the annotation, answering "Yes" or "No".

**Cross-Scene Consistency.**

- **Character.** Characters appearing at different timestamps are expected to maintain consistent visual characteristics. To evaluate this, we select images that should depict the same character according to the ground-truth annotations and ask the judge whether these images contain the same character with consistent attributes. This process is repeated for all characters, and we report the ratio of positive responses for each storyboard.

- **Location.** Similar to character consistency, we ask the judge whether multiple generated images are taken at the same location and report the corresponding positive response ratio for each storyboard.

- **Style.** We provide all generated images within a storyboard and ask the judge whether they share a consistent visual style. The judge responds "Yes" or "No".

**VLM Judge Prompt for Character Number Alignment**

## System Prompt
You are a helpful and objective judge. You must be strict, consistent, and unbiased, and rely only on observable evidence from the inputs.

## Instruction
<image>
How many characters are in this image? Only answer an Arabic number.

---

**VLM Judge Prompt for Character Attributes Alignment**

## System Prompt
You are a helpful and objective judge. You must be strict, consistent, and unbiased, and rely only on observable evidence from the inputs.

## Instruction
<image>
<First Character Name>: <Description>
<Second Character Name>: <Description>
...
Do characters in this image fit into their descriptions? Only answer yes or no.

---

**VLM Judge Prompt for Non-Character Entities Alignment**

## System Prompt
You are a helpful and objective judge. You must be strict, consistent, and unbiased, and rely only on observable evidence from the inputs.

## Instruction
<image>
Does this image contain or imply <entity #1>? Only answer yes or no.
Does this image contain or imply <entity #2>? Only answer yes or no.
...

---

**VLM Judge Prompt for Location Alignment**

## System Prompt
You are a helpful and objective judge. You must be strict, consistent, and unbiased, and rely only on observable evidence from the inputs.

## Instruction
<image>
Is this image taken at a/an <location>? Only answer yes or no.

---

**VLM Judge Prompt for Time Alignment**

## System Prompt
You are a helpful and objective judge. You must be strict, consistent, and unbiased, and rely only on observable evidence from the inputs.

## Instruction
<image>
Is this image taken at/in the <time>? Only answer yes or no.

---

**VLM Judge Prompt for Character Consistency**

## System Prompt
You are a helpful and objective judge. You must be strict, consistent, and unbiased, and rely only on observable evidence from the inputs.

## Instruction
<Selected image #1 that should contain the First Character>
<Selected image #2 that should contain the First Character>
...
Do all these images contain the same character <First Character Name>: <Description>? Only answer yes or no.

<Selected image #1 that should contain the Second Character>
<Selected image #2 that should contain the Second Character>
...
Do all these images contain the same character <Second Character Name>: <Description>? Only answer yes or no.
...

---

**VLM Judge Prompt for Location Consistency**

## System Prompt
You are a helpful and objective judge. You must be strict, consistent, and unbiased, and rely only on observable evidence from the inputs.

## Instruction
<Selected image #1 that should be taken at the First Location>
<Selected image #2 that should be taken at the First Location>
...
Are all these images taken at the same <First Location>? Only answer yes or no.

<Selected image #1 that should be taken at the Second Location>
<Selected image #2 that should be taken at the Second Location>
...
Are all these images taken at the same <Second Location>? Only answer yes or no.
...

---

**VLM Judge Prompt for Style Consistency**

## System Prompt
You are a helpful and objective judge. You must be strict, consistent, and unbiased, and rely only on observable evidence from the inputs.

## Instruction
<image #1>
<image #2>
<image #3>
...
Are all these images in the same style? Only answer yes or no.

---

### B.4. Benchmark Examples

In Figure 6, we showcase several example storyboards created by our model, DynaVieW and BAGEL after SFT, along with ground-truth storyboards. In Figure 7, we present 4 examples sampled from the Epic-Kitchen dataset.

## B.5. Supervised Fine-tuning Details

For supervised fine-tuning on each downstream benchmark (VinaBench or LEGO), we further fine-tune DynaVieW (and baseline models) on the official training samples provided by the benchmark, including 11652 visual narrative samples from the Visual Writing Prompts (VWP) portion of VinaBench, or 147362 world simulation samples from LEGO, respectively. We fine-tune models on a cosine learning rate scheduler, with warm-up steps, total training steps, maximum learning rate and minimum learning rate set to 300, 6000, $5e^{-6}$ and $1e^{-7}$, respectively. We deploy the training on 32 NVIDIA GH200 GPUs, which takes about 2.5 days. Same as the pre-training, we freeze the weights of the VAE encoder during the fine-tuning of DynaVieW, and use AdamW (Loshchilov & Hutter, 2018) optimizer with $\beta_1$, $\beta_2$ and $\epsilon$ set to 0.9, 0.95 and $1e^{-15}$, respectively. The maximum norm for gradient clipping also remains to be 1.0.

## B.6. Inference Settings

We systematically compare our model with baseline methods across multiple inference settings, including zero-shot, supervised fine-tuning, and model-assisted transition generation for the controllability experiment, where the transition format differs across settings.

**Zero-shot** Only the ground-truth transition narrative is included in the interleaved input stream.

**Supervised fine-tuning** In addition to the ground-truth transition narrative, we prompt both our model and the SFT baseline to generate a JSON schema in the style of VinaBench annotations, which is then added to the interleaved input stream.

**Model-assisted transition generation** In addition to the ground-truth transition narrative, we use Gemma-3 (27B-it) to generate a JSON schema matching the format of the pre-training data, which is then added to the interleaved input stream. For a fair comparison with BEGAL, we also convert the JSON schema into a natural-language representation as a variant.

We admit that Gemma-3 (27B-it) has not been broadly pre-trained on world modelling tasks to understand the visual dynamics, but it has strong instruction-following performance. Therefore, we enhance its capability of visual dynamics understanding through sophisticated prompts, to fine-grainedly instruct its transition prediction. Moreover, we expect Gemma to create diverse (more important than accurate in controllability experiments) transitions to test our DynaVieW model's controllability of state simulation based on varied possible transitions, and noisy transitions can be used to examine the robustness (reflected by the low variance of scores shown in Table 3) of our model in resisting potentially infeasible state simulation conditions.

---

**Zero-Shot Transition Template**

<Textual Narrative>: Danny has come back home from his trip in Hawaii, and is greeting all of his neighbours.

---

**Supervised Fine-Tuning Transition Template (VinaBench)**

<Textual Narrative>: Danny has come back home from his trip in Hawaii, and is greeting all of his neighbours.
<Predicted Schema>:
```
{
    "style": "photorealistic",
    "entities": [
        "someone is arriving",
        "someone",
        "home",
        "vacation"
    ],
    "characters": ["Danny", "neighbours"],
    "time": "afternoon",
    "location": "house",
    "caption": "..."
}
```

**Gemma3-generated Natural Language Transition Template**

<Textual Narrative>: Danny has come back home from his trip in Hawaii, and is greeting all of his neighbours.
<Predicted Schema>:
High-level activity:
The overall activity depicted is Greeting Neighbors.

Sub-activity: Person approaching and greeting
This sub-activity focuses on an individual moving toward nearby neighbors and initiating a greeting.

Atomic action: Danny approaching neighbors
In this atomic action, Danny is introduced as an active participant in the scene. The surrounding environment remains unchanged and continues to include a house, street, plants, and a car, all of which persist throughout the action.

Danny's spatial position changes: he moves from being on or near the house (previously associated with the roof area) to a location closer to the neighbors. Although this movement is not directly visible in the current frame, it is inferred from the context. There are no changes in object states such as color, shape, physical condition, or texture, and no new physical contact or alignment relationships are formed.

The action is characterized by initiating behaviors, specifically walking and greeting. Correspondingly, Danny exhibits translational motion as he walks toward the neighbors, while no rotational, oscillatory, or deformational motion is involved. The camera remains static, with no changes in viewpoint, zoom, pan, tilt, or focus. Likewise, the background—lighting, atmosphere, scene context, and overall environment—remains constant.

Contribution and significance:
   • At the atomic level, the introduction of Danny and his inferred movement toward the neighbors directly represents the act of approaching and greeting them.
   • At the sub-activity level, this action fulfills the central purpose of the "Person approaching and greeting" sequence.
   • Temporally, this atomic action serves as the initiating event, setting up the subsequent interaction between Danny and the neighbors.

**Gemma3-generated JSON-Schema Transition Template**

<Textual Narrative>: Danny has come back home from his trip in Hawaii, and is greeting all of his neighbours.
<Predicted Schema>:
```
{
    "high_level_activity": "Greeting Neighbors",
    "sub_activities": [ {
        "name": "Person approaching and greeting",
        "atomic_actions": {
            "name": "Danny approaching neighbors",
            "transitions": {
                "objects": {
                    "introduced": [
                        "Danny"
                    ],
                    "persistent": [
                        "House",
                        "Street",
                        "Plants",
                        "Car"
                    ]
                },
                "object_states": {...}
                "spatial_relations": {
                    "positional_changes": [
                        "Danny's position changes from being on the roof/near the house to presumably near the neighbors,
though not visible in the current frame."
                    ],
                    ...
                },
                "actions": {
                    "initiating": [
                        "Walking",
                        "Greeting"
                    ],
                    ...
                },
                "motion": {
                    "translational": [
                        "Danny exhibits translational motion as he walks towards the neighbors (inferred)"
                    ],
                    ...
                },
                "camera": {...}
                "background": {...}
            },
            "contributions": {
                "to_atomic_action": "The introduction of 'Danny' into the scene (though his initial position is not
visible here) is directly linked to him starting to approach and greets his neighbors.",
                "to_sub_activity": "This action fulfills the core goal of the 'Person approaching and greeting' sub-
activity.",
                "temporal_significance": "This is the initiating action that sets the stage for showcasing the arrival and
subsequent interaction with neighbors."
} } ] } ] }
```

*Table 6.* Evaluation results of visual narrative generation on VinaBench, with Gemini-2.5-Pro as the judge. "Non-Char. Ent.", "Char. Num." and "Char. Attr." indicate Non-Character Entities, Character Number and Character Attribute, respectively. The best results within each block are **in bold** and the second-best are underlined.

| Model | Per-Scene Alignment | | | | | Cross-Scene Consistency | | | Average |
|---|---|---|---|---|---|---|---|---|---|
| | Non-Char. Ent. | Char. Num. | Char. Attr. | Time | Location | Style | Character | Location | |
| **Zero-Shot Generalization Performance** | | | | | | | | | |
| BAGEL | 0.720 | 0.280 | **0.533** | 0.626 | 0.537 | 0.557 | 0.112 | 0.323 | 0.435 |
| Story2Board | **0.723** | **0.339** | 0.428 | 0.369 | 0.411 | **0.815** | 0.280 | 0.225 | 0.447 |
| DynaVieW | 0.719 | 0.278 | 0.527 | **0.655** | **0.556** | 0.645 | **0.284** | **0.610** | **0.530** |
| **Supervised Fine-Tuning** | | | | | | | | | |
| ARLDM | 0.765 | 0.360 | 0.576 | 0.469 | 0.465 | 0.397 | 0.061 | 0.097 | 0.356 |
| MM-Interleaved | 0.770 | 0.375 | 0.581 | 0.521 | 0.499 | **0.650** | 0.081 | 0.152 | 0.422 |
| StoryGen | 0.738 | 0.370 | 0.440 | 0.367 | 0.412 | 0.131 | 0.048 | 0.055 | 0.272 |
| BAGEL | 0.760 | **0.392** | **0.585** | 0.575 | **0.547** | 0.569 | 0.058 | 0.168 | 0.418 |
| DynaVieW | **0.777** | 0.371 | 0.583 | **0.601** | 0.541 | 0.643 | **0.082** | **0.239** | **0.448** |

*Table 7.* Image-to-text metrics of our model and baselines on LEGO. The best results within each block are **in bold**.

| Model | Ego4D | | Epic-Kitchens | |
|---|---|---|---|---|
| | BLIP-B | BLIP-L | BLIP-B | BLIP-L |
| LEGO | 20.38 | 20.70 | 26.98 | 27.41 |
| BAGEL | 23.01 | 22.74 | 26.02 | 26.60 |
| DynaVieW | **25.02** | **24.90** | **27.27** | **27.93** |

# C. More Experimental Results

## C.1. The Trade-off Between Per-Scene Alignment and Cross-Scene Consistency

The experimental results in Table 2 indicate that supervised fine-tuning on downstream data improves per-scene alignment (0.566 vs 0.499), but at the cost of reduced cross-scene consistency (0.654 vs 0.761). We attribute this trade-off to the tendency of supervised fine-tuning to encourage close imitation of ground-truth scenes, including detailed transitions and visual manifestations, which leads to higher per-scene alignment scores. However, as illustrated by the storyboard examples in Figure 6, this overemphasis on per-scene alignment promotes richer local transformations while weakening the visual coherence between consecutive scenes, ultimately degrading long-term consistency.

## C.2. Gemini-2.5-Pro As the Judge

Table 6 compares visual narrative generation performance under both zero-shot and supervised fine-tuning settings using Gemini-2.5-Pro as the judge. In the zero-shot regime, DynaVieW achieves the highest overall average score (0.530), substantially outperforming BAGEL (0.435) and Story2Board (0.447). Under supervised fine-tuning, DynaVieW continues to demonstrate robust performance, achieving the highest average score (0.448) among all methods. While BAGEL attains slightly better per-scene alignment scores on character number, attributes, and location, DynaVieW consistently outperforms alternatives on cross-scene consistency metrics, particularly for character (0.082) and location (0.239). Overall, the additional evaluations with Gemini-2.5-Pro as the judge further demonstrate our model's strong world-modeling capability.

## C.3. Image-to-Text Metrics on LEGO

We report results of our model and baselines under six image-to-image metrics on LEGO in Table 4, which are based on comparisons between the output images and ground-truth images. Prior work (Lai et al., 2024) shows that the widely-used image-to-text CLIP score cannot align actions with egocentric images due to the domain gap. So we implement another two image-to-text metrics, BLIP-B and BLIP-L, where we generate captions for the output images using two vision-language pre-trained BLIP (Li et al., 2022) models and calculate the text-to-text similarity score between the captions and action instructions. We report the results of BLIP-B and BLIP-L in Table 7.

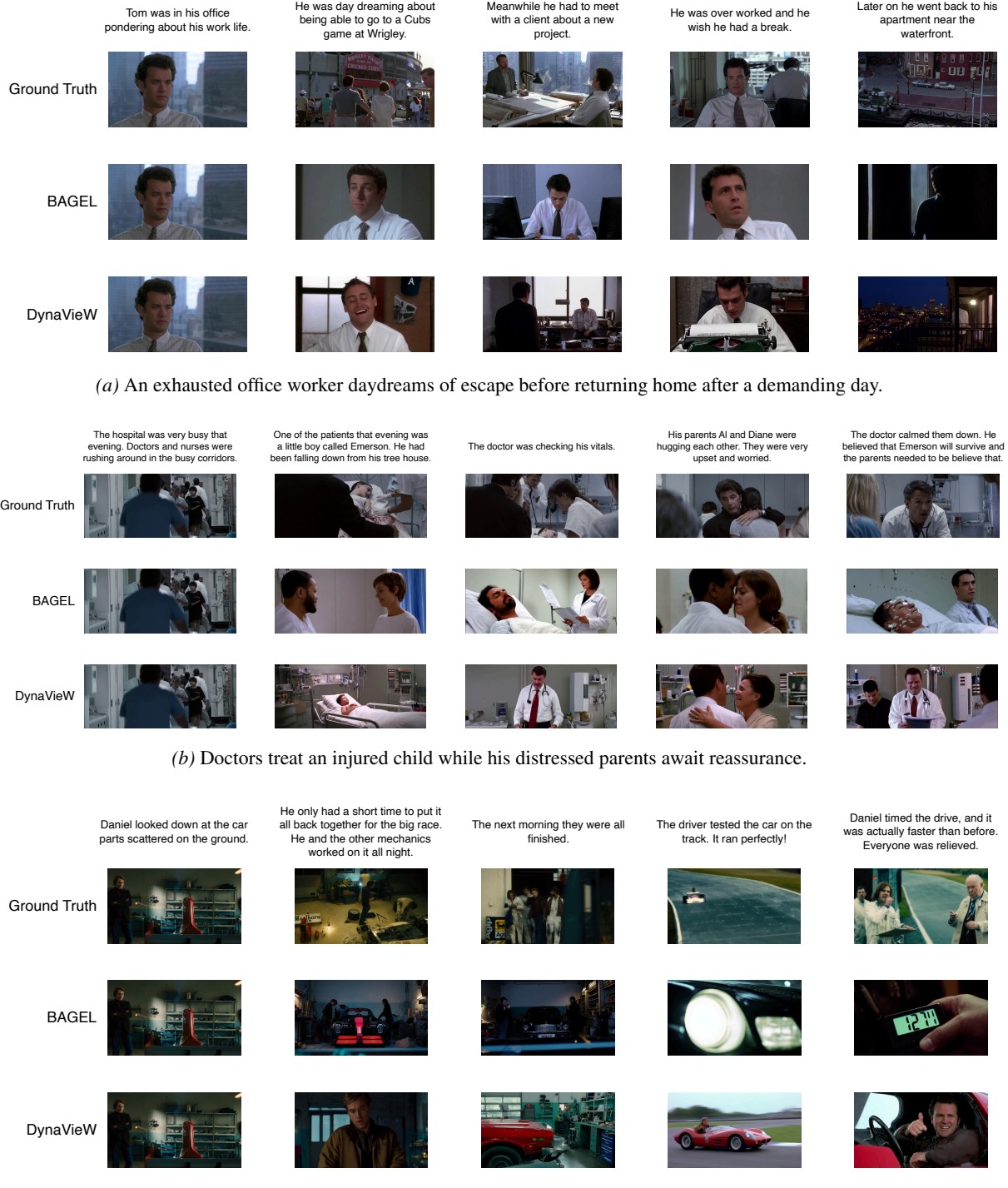

*(a)* An exhausted office worker daydreams of escape before returning home after a demanding day.

*(b)* Doctors treat an injured child while his distressed parents await reassurance.

*(c)* A race car is rebuilt overnight and proves its improved performance during testing.

*Figure 6.* Example storyboards generated by DynaVieW and BAGEL after SFT, along with ground truth storyboards from VinaBench.

| Input Frame | Ground Truth | BAGEL | DynaVieW |
|---|---|---|---|

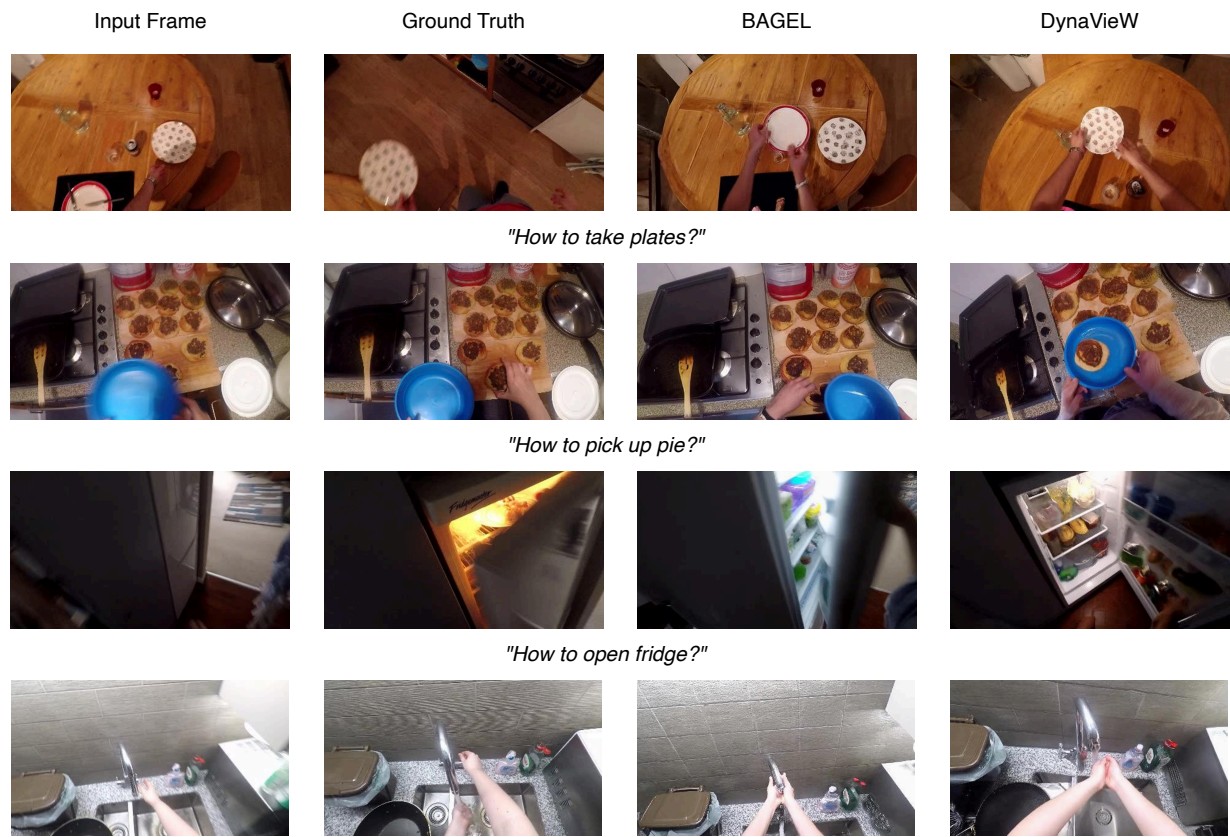

*"How to take plates?"*

*"How to pick up pie?"*

*"How to open fridge?"*

*"How to wash hands?"*

*Figure 7.* World simulation outputs generated by DynaVieW and BAGEL after SFT, along with ground truth visual answers from LEGO.

## C.4. Compute Cost and Latency of DynaVieW

We have tested the compute and latency costs of models on the downstream visual narrative generation task using a single NVIDIA H100 GPU, averaging results over 300 random testing samples from VinaBench dataset. The latency (in seconds) and the approximated KV-Cache (in MB) consumed for generating a visual narrative state with respect to the number of its preceding (previously generated) states in the context is shown in Table 8. DynaVieW and BAGEL do state-only predictions.

We acknowledge that the maximum length of visual sequences that can be handled by our DynaVieW model is limited to the backbone model (BAGEL) capacity and computing, which only support modeling a maximum number of 36,864 tokens. And the evaluation on VinaBench, which only consists of short storyboards (e.g., 5-9 visual states), is constrained. Scaling up to longer sequences requires a larger token budget and is out of our paper scope, which we leave for future work.

*Table 8.* Latency and compute cost of DynaVieW and BAGEL under different numbers of preceding states

| # Preceding States | Latency (s) | | Compute Cost (MB) | |
|---|---|---|---|---|
| | DynaVieW | BAGEL | DynaVieW | BAGEL |
| 1 | $17.92 \pm 0.77$ | $18.15 \pm 0.76$ | $407.71 \pm 19.71$ | $406.77 \pm 20.66$ |
| 2 | $18.50 \pm 0.89$ | $18.74 \pm 0.85$ | $502.91 \pm 28.75$ | $501.38 \pm 30.10$ |
| 3 | $19.11 \pm 1.01$ | $19.32 \pm 1.00$ | $582.73 \pm 35.10$ | $580.32 \pm 35.30$ |
| 4 | $19.72 \pm 1.12$ | $19.96 \pm 1.10$ | $646.08 \pm 49.59$ | $645.85 \pm 48.35$ |
| 5 | $20.30 \pm 1.22$ | $20.55 \pm 1.22$ | $724.30 \pm 61.51$ | $722.40 \pm 62.08$ |

## C.5. More Baselines on LEGO

We add more baselines from prior work (Lai et al., 2024) on LEGO benchmark. The evaluation results in Table 9 (tested on the Epic-Kitchen portion) show that the performance of DynaVieW is consistently better than other (SFT) baselines.

*Table 9.* More Baselines on LEGO Benchmark

| | FID ↓ | PSNR ↑ | LPIPS ↓ | CLIP ↑ | EgoVLP ↑ | EgoVLP$^{+}$ ↑ |
|---|---|---|---|---|---|---|
| ProxEdit (Han et al., 2023) | 51.35 | 11.06 | 46.35 | 65.80 | 32.27 | 52.77 |
| SDEdit (Meng et al., 2021) | 27.41 | 11.30 | 43.33 | 74.76 | 33.84 | 56.80 |
| IP2P (Brooks et al., 2023) | 20.64 | 11.23 | 40.82 | 77.03 | 42.97 | 61.06 |
| LEGO (Lai et al., 2024) | 21.57 | 11.33 | 40.36 | 78.63 | 45.89 | **62.66** |
| DynaVieW (Zero-Shot) | 18.33 | **11.48** | 41.78 | 77.79 | 43.60 | 58.83 |
| DynaVieW (SFT) | **10.31** | 11.31 | **39.20** | **84.19** | **48.72** | 61.77 |

