# OpenReview forum: "DynaVieW: Schema-Guided World Modeling for Understanding Hierarchical Visual Dynamics"
_ICML.cc/2026/Conference — ICML 2026 regular_

### Official Review · Reviewer_Xq8w · 2026-03-03

**Soundness:** 2
**Presentation:** 3
**Significance:** 3
**Originality:** 3
**Overall Recommendation:** 4
**Confidence:** 3

**Summary:**

To address the current limitations of MLLMs in capturing hierarchical visual changes, the authors transform world modeling into learning interleaved state-transition sequences. "States" refer to video keyframes, while "transitions" are represented by a highly structured, hierarchical JSON schema that covers high-level activities as well as fine-grained visual transformations (e.g., object states, spatial relationships, and camera movements).

DynaVieW employs a Mixture-of-Transformer (MoT) framework, integrating a visual encoder (SigLIP2) and a VAE encoder (FLUX) with a shared LLM backbone (Qwen2.5-7B). To facilitate training, the authors introduce a multimodal selective attention mechanism to prevent information leakage and propose a schema token re-weighted cross-entropy (CE) loss to balance the learning between JSON structural tokens and semantic slot values.

Annotated using InternVL-78B-Instruct, the model was pre-trained on a diverse dataset constructed from Ego4D, AgiBotWorld-Alpha, and ShareGPT4Video. Empirical evaluations on VinaBench and LEGO demonstrate that DynaVieW achieves superior performance in cross-scene consistency, controllability, and visual fidelity compared to strong baselines such as BAGEL and Story2Board.

**Compliance With Llm Reviewing Policy:**

Affirmed.

**Final Justification:**

I appreciate this paper's innovative attempt to use a hierarchical JSON schema to represent state transitions for world modeling. The authors' response during the rebuttal phase has also deepened my understanding of this work. This reassures me and I will maintain my original positive score.

**Key Questions For Authors:**

Compared to baselines like BAGEL, can the token overhead and inference latency of generating a full JSON schema between each frame be quantified? Has the exploration of dynamic schema truncation (e.g., completely dropping empty fields) been considered to save tokens during inference?

Regarding error accumulation in long sequences: if an incorrect JSON transition is generated at step $t=1$, to what extent does it degrade the FID/CLIP scores by step $t=5$?

Could a human evaluation (e.g., Elo rating or side-by-side preference comparison) be conducted to corroborate the VQA-based metrics provided by the LLM judges?

**Limitations:**

As mentioned in the implementation details, due to memory constraints, the model is required to split sequences into windows containing 6 states and 5 transitions. This inherently limits the model's ability to natively attend to very long-term dependencies (e.g., referencing a visual state from 20 steps prior) without relying strictly on recent textual transitions.

The strictly defined JSON schema, while excellent for structured learning, may struggle to describe abstract, surreal, or highly unconventional dynamic transitions that do not fit neatly into predefined "atomic actions" or "object state transformations."

**Strengths And Weaknesses:**

**Strengths**

The adaptation of the MoT architecture is well-executed. The introduction of the schema token re-weighted CE loss is a highly practical and clever solution; it prevents the model from overfitting on the repetitive structural tokens of the JSON format while avoiding under-learning of the actual dynamic values.

The evaluation covers both open-ended, long-horizon generation (Visual Storytelling via VinaBench) and deterministic instruction following (World Simulation via LEGO). Evaluations under both zero-shot generalization and SFT settings provide a comprehensive demonstration of the model's capabilities.

**Weaknesses**

Relying on the generation of an exhaustive, highly detailed JSON schema between every visual state introduces significant token overhead. Compared to models that predict raw pixel space or use compressed latent transitions, this could drastically increase inference latency and computational costs.

While using GPT-4o and Gemini-2.5-Pro as judges for VinaBench is a modern standard, visual storytelling quality and spatio-temporal consistency are inherently subjective. The lack of human preference studies weakens the persuasiveness of the claims regarding visual quality and alignment.

In long-horizon autoregressive generation, if the transition expert hallucinates structural changes within the JSON schema or predicts incorrect underlying transformations, it remains unclear whether the visual generation expert can recover, or if this leads to a cascading failure in subsequent frames.

The quality of the entire pre-training dataset is limited by the capabilities of InternVL-78B-Instruct. Any systematic biases or blind spots this "Oracle" model has regarding physical commonsense will be directly baked into DynaVieW.

---

> ### Author Rebuttal · Authors · 2026-03-30
>
> We thank the reviewer for recognizing that our world modeling method is well-executed, with a highly practical and effective schema token re-weighted training loss, and for acknowledging that our evaluation results are comprehensive.
>
> **(W1 and Q1)** The reviewer mentions that generating detailed JSON schemas of transitions between visual states could increase our model’s inference costs, and suggests using a dynamic schema truncation to save inference tokens.
>
> We have tested the compute and latency costs of models on the downstream visual narrative state generation task using a single NVIDIA H100 GPU, averaging results over 300 testing samples. Results are shown in the following table, where DynaVieW and BAGEL do state-only prediction (i.e., zero-shot generalization in Table 2), and DynaVieW + Schema additionally predicts the JSON-schema transition before simulating the next state. Avg state latency measures the average time (in seconds) of generating a single state image.
>
> |Metric|DynaVieW|BAGEL|DynaVieW+Schema|
> |-|-|-|-|
> |Avg state latency (s)|19.06 ± 1.30|19.30 ±1.30|31.62 ± 5.85|
>
> Indeed, generating hierarchical JSON transitions at inference time adds latency as shown in the table above, increasing 65.9% per-state inference latency on average. We will add this limitation to the paper.
>
> We also thank the reviewer for the suggestion of dynamic schema truncation, and have estimated that dropping empty fields in the schema would save about 60% of transition tokens at inference time. However, such a method is out of our paper’s current scope, and we leave it as a future research direction.
>
> **(W2 & Q3)** The reviewer asks for a human evaluation to further corroborate the VQA-based metrics based on the LLM judges, which we used for evaluating the downstream visual narrative generation on the VinaBench dataset.
>
> Following the reviewer’s suggestion, we conduct a head-to-head human evaluation between DynaVieW (SFT) and BAGEL (SFT). For each storyboard example, we ask a human tester to read a short list of textual narratives that describe a film scene, and then compare two storyboards generated by both models (anonymously presented and randomly ordered). In total, we collect 400 preference votes on which storyboard better matches the narrative, 50 votes for each human tester. The results show that human testers prefer DynaVieW in 47.6% of cases, compared to 35.4% for BAGEL, with 17% of comparisons resulting in a tie. The human evaluation results align well with automatic VLM judges, clearly showing an advantage for DynaVieW compared to BAGEL.
>
>
> **(W3 and Q2)** The reviewer concerns whether hallucinations or errors in the predicted transitions would propagate to affect the simulation of subsequent visual states.
>
> We highlight that the multimodal selective attention used in our DynaVieW model mitigates error propagation across the state-transition sequence generation as the simulation of visual state (at step t) only draws attention to its last preceding transition, and not to more previously predicted transitions (from step 0 to step t-1). This ensures that transition prediction hallucinations or errors in the more previous time steps do not directly affect the current (t’s) state simulation. However, we acknowledge that error propagation may still exist implicitly across the simulated states, which is however a general challenge for any image sequence generation task.
>
> **(W4)** The reviewer concerns that the quality of our world model pre-training data is limited by the capabilities of the oracle VLM (InternVL-78B-Instruct), which annotates the core JSON schemas of state transitions.
>
> In Table 1, we verified that the gold annotations of the oracle VLM achieved a low (< 2%) reject rate when evaluated by the judge (GPT-4o). We also manually annotated 30 transitions to verify that the oracle VLM’s annotations are well-aligned with our human annotations, with an average alignment score around 8, on a Likert scale from 1 to 10 (higher is better). We acknowledge that the Oracle may still have blind spots outside this holistic evaluation, but our downstream results demonstrate that DynaView still learns usable transition dynamics from these annotations.
>
> **Limitations:** We thank the reviewer for pointing out the above two limitations and will include the related discussions in our camera-ready version.

---

> > ### Author Rebuttal · Reviewer_Xq8w · 2026-04-01
> >
> > Thank you to the authors for their detailed and thoughtful responses. All of my concerns have been satisfactorily addressed. I will be maintaining my original score.

---

### Official Review · Reviewer_8gir · 2026-03-08

**Soundness:** 3
**Presentation:** 1
**Significance:** 3
**Originality:** 3
**Overall Recommendation:** 4
**Confidence:** 4

**Summary:**

- DynaVieW is a dynamic visual schema-guided world model that addresses the challenge of understanding hierarchical visual dynamics in multimodal LLMs. The core idea is to pre-train on interleaved state-transition sequences, where visual states are keyframes from diverse real-world videos (Ego4D, AgiBotWorld-Alpha, ShareGPT4Video), and transitions are hierarchical JSON-schema texts generated by an oracle VLM (InternVL-78B-Instruct).

- The transition schema captures a rich hierarchy: high-level activities, sub-activities, atomic actions, seven types of transformations (objects, spatial relations, actions, motion, camera, background), and three types of contribution analyses.

- The model builds on BAGEL's Mixture-of-Transformer-Experts (MoT) architecture and introduces two key training innovations: (1) a cross-expert selective attention mask that prevents naive copying of historical transitions and reduces redundancy, and (2) a schema token re-weighted cross-entropy loss that down-weights static JSON schema tokens (weight 0.1) to focus learning on slot-filling content tokens. DynaVieW is evaluated on visual narrative generation (VinaBench) and world simulation (LEGO), demonstrating consistent gains over BAGEL and other baselines in both zero-shot and supervised fine-tuning settings.

**Compliance With Llm Reviewing Policy:**

Affirmed.

**Final Justification:**

Thank you for your rebuttal. I will be maintaining my original score.

**Key Questions For Authors:**

- Ablation: Can you provide results with (a) DynaVieW trained without the selective attention mask, (b) DynaVieW with uniform CE loss (no schema token re-weighting), and (c) DynaVieW trained with flat natural-language transitions instead of JSON schema? This would greatly strengthen the paper's technical claims.

**Limitations:**

Yes

**Strengths And Weaknesses:**

**Strength**
- The 7-type transformation taxonomy (objects, object states, spatial relations, actions, motion, camera, background) + 3-level activity hierarchy is thorough and grounded in visual cognition literature.
- The two-stage keyframe extraction (sharpness via Laplacian variance + CLIP cosine similarity threshold) is well-motivated and avoids redundant or blurry frames.
- The decision to mask transition-to-transition attention (preventing naive copying) while preserving state ViT encodings as implicit historical context is an elegant solution to the sequence modeling challenge.

**Weaknesses**
- **The paper's most significant missing piece is an ablation study.** The three main technical contributions -- (a) the hierarchical schema vs. flat description, (b) the selective attention mask, and (c) the schema token re-weighted loss -- are never isolated.
- Without ablations, it is unclear how much each component contributes to the overall improvement. For example: would a flat natural-language transition description (without JSON schema) achieve similar gains? The controllability experiment (Table 3) hints at the schema's importance, but does not isolate the model's pre-training objective from its input format.
- Specifically, DynaVieW-NL vs. BAGEL-NL in Table 3 shows that DynaVieW's advantage persists even with natural language input, which could mean the selective attention or re-weighted loss is the primary driver -- but this is not directly tested.
- In the VinaBench evaluation (Table 2, SFT), the paper only compares against BAGEL and older autoregressive diffusion baselines (ARLDM, MM-Interleaved, StoryGen). There is no comparison with recent strong multimodal generation models such as Show-1, Emu2, or other state-of-the-art interleaved LLMs. The LEGO benchmark only compares DynaVieW against BAGEL. A single baseline is insufficient to establish general superiority.
- The entire transition annotation pipeline depends on InternVL-78B-Instruct, which is a 78B VLM. This creates two concerns: (a) the annotation quality is bounded by InternVL's visual understanding, and (b) the computational cost of annotating new data at scale is very high.
- Generating hierarchical JSON transitions at inference time (especially in the SFT controllability setting) adds substantial latency. The paper does not discuss inference time or the practical cost of DynaVieW's approach relative to BAGEL.

---

> ### Author Rebuttal · Authors · 2026-03-30
>
> We thank the reviewer for recognizing that our proposed world modeling method is sound and well-framed, including carefully-filtered visual state extraction, thorough state transition schema, and elegant multimodal selective attention.
>
> **1. (Ablation Study)** The reviewer asks for an ablation study on the technical contributions of our DynaVieW model.
>
> We conduct a detailed ablation study on each major component of our method, including: (a) removing the multimodal selective attention (**w/o Selective Attn.**), (b) removing the schema token re-weighting of CE loss (**w/o Re-weighted Loss**), (c) using a heuristic template to translate the JSON-schema transitions into natural language descriptions (**w/o JSON-Schema**).
>
> We compare the performances of our original DynaVieW and the ablated models on the downstream VinaBench dataset, under the zero-shot generalization setting used in Table 2.
>
> |Model|Non-Char. Ent.|Char. Num.|Char. Attr.|Time|Location (Align.)|Style|Character|Location (Cons.)|Average|
> |-|-|-|-|-|-|-|-|-|-|
> |DynaVieW|0.600|0.323|0.521|0.551|0.501|0.897|0.550|0.835|0.630|
> |w/o Selective Attn.|-0.018|-0.021|-0.019|-0.020|-0.016|-0.023|-0.007|+0.011|-0.013|
> |w/o Re-weighted Loss|-0.005|-0.011|-0.005|-0.007|-0.003|-0.024|+0.019|+0.003|-0.004|
> |w/o JSON-Schema|-0.019|-0.009|-0.013|-0.009|-0.010|-0.067|-0.031|-0.029|-0.028|
>
> Results above show that removing the multimodal selective attention consistently makes the model perform worse. Without the schema token re-weighted CE loss, the model also underperforms DynaVieW, although with a smaller performance margin. Our ablation on the schema further shows that JSON-structured descriptions of visual dynamics are necessary for the success of DynaVieW training. We will include these ablation study results in our camera-ready version.
>
> **2. (Baselines)** The reviewer asks to compare our DynaVieW model to more baseline models in our downstream evaluations on VinaBench and LEGO datasets.
>
> On VinaBench, we tested Emu2 under the zero-shot generalization setting, to complement our evaluation results in Table 2. DynaVieW (7B) outperforms Emu2 (37B) consistently across all metrics except Character Attributes, with particularly notable advantage in cross-scene consistency metrics.
>
> |Model|Non-Char. Ent.|Char. Num.|Char. Attr.|Time|Location (Align.)|Style|Character| Location (Cons.) | Average |
> |-|-|-|-|-|-|-|-|-|-|
> |Emu2|0.533|0.220|0.556|0.523|0.466|0.378|0.352|0.439|0.425|
> |DynaVieW|0.600|0.323|0.521|0.551|0.501|0.897|0.550|0.835|0.630|
>
> We did not include Show-1 because this model is optimized for generating video clips instead of visual narratives (i.e., storyboard images). We also tested other interleaved image-text generative models, e.g., Anole and Lumina-mGPT, which completely failed (with nearly zero scores) under the zero-shot generalization setting.
>
> We also add more baselines from prior work on LEGO benchmark. The evaluation results in the following table (tested on the Epic-Kitchen portion) show that the performance of DynaVieW is consistently better than various (SFT) baselines.
>
> |Model|FID ↓|PSNR ↑|LPIPS ↓|CLIP ↑|EgoVLP ↑|EgoVLP+ ↑|
> |-|-|-|-|-|-|-|
> |ProxEdit|51.35|11.06|46.35|65.80|32.27|52.77|
> |SDEdit|27.41|11.30|43.33|74.76|33.84|56.80|
> |IP2P|20.64|11.23|40.82|77.03|42.97|61.06|
> |LEGO|21.57|11.33|40.36|78.63|45.89|**62.66**|
> |DynaVieW (Zero-Shot)|18.33|**11.48**|41.78|77.79|43.60|58.83|
> |DynaVieW (SFT)|**10.31**|11.31|**39.20**|**84.19**|**48.72**|61.77|
>
> **3. (Transition Annotation)** The reviewer concerns that the transition annotation quality of the oracle VLM (InternVL-78B-Instruct) will be bounded by its visual understanding, and the computational cost of annotating new data at scale will be very high.
>
> In Table 1, we verified that the gold annotations of the oracle VLM achieved a low (< 2%) reject rate when evaluated by the judge (GPT-4o). We also manually annotated 30 transitions to verify that the oracle VLM’s annotations are well-aligned with our human annotations, with an average alignment score around 8, on a Likert scale from 1 to 10 (higher is better). We acknowledge that the Oracle may still have blind spots outside this holistic evaluation, but our downstream results demonstrate that DynaView still learns usable transition dynamics from these annotations.
>
> About the cost of annotating new data, we highlight that as shown by the transition validation results in Table 1, our pre-trained DynaVieW model already achieves a transition prediction performance that is close to the gold oracle. This indicates that DynaVieW can potentially serve as a reliable and lighter transition annotator to more efficiently annotate new data at scale, instead of still prompting the larger oracle VLM.
>
> **4. (Latency of Generation)** The reviewer concerns that generating our hierarchical JSON-schema transitions at inference time will add substantial latency.
>
> We include our measure of latency and discussions in our response to **Reviewer Xq8w (W1 and Q1)**.

---

### Official Review · Reviewer_AXi5 · 2026-03-12

**Soundness:** 2
**Presentation:** 3
**Significance:** 3
**Originality:** 2
**Overall Recommendation:** 4
**Confidence:** 3

**Summary:**

Proposes DynaView, a schema-guided world model that learns interleaved state-transition sequences from video keyframes, where transitions are hierarchical JSON descriptions of dynamics. The model jointly predicts transitions and simulates states with a mixture-of-experts architecture, selective attention, and schema token reweighting. Experiments on visual narrative generation (VinaBench) and world simulation (LEGO) show improved consistency, controllability, and instruction following compared to baselines.

**Compliance With Llm Reviewing Policy:**

Affirmed.

**Key Questions For Authors:**

1) How sensitive are results to schema granularity and the schema token reweighting loss?
2) What is the measured annotation noise level for transitions, and how robust is the model to noisy schemas?
3) At inference, can the model work with free-form prompts, or does it require the full JSON schema?
4) What are the compute and latency costs for long sequences (e.g., 50-100 states), and how do they scale?
5) Do improvements hold on truly long-horizon videos outside the pretraining sources?

**Limitations:**

Not fully; please discuss reliance on VLM-generated annotations, potential dataset bias, and risks of misuse for realistic video synthesis.

**Strengths And Weaknesses:**

Strengths:
1) Clear motivation and a well-structured hierarchical schema for capturing multi-level dynamics.
2) Joint modeling of transition prediction and state simulation with MoT is novel and well motivated.
3) Broad training data from diverse video sources and strong benchmark results with supporting ablations.

Weaknesses:
1) Relies on VLM-generated transition annotations; noise and bias are not fully quantified.
2) Evaluation relies heavily on automatic/VLM judges with limited human evaluation and failure analysis.
3) Inference cost and scalability to long sequences are not fully characterized.

---

> ### Author Rebuttal · Authors · 2026-03-30
>
> We thank the reviewer for acknowledging that our proposed state-transition world modeling method is novel and well-motivated, and for recognizing our broad world model pre-training data and strong benchmark results.
>
> **(W1 and Q2)** The reviewer asks for quantifying the noise and bias in the gold transitions annotated by the oracle VLM (InternVL-78B-Instruct).
>
> In Table 1, we verified that the gold annotations of the oracle VLM achieved a low (< 2%) reject rate when evaluated by the judge (GPT-4o). We also manually annotated 30 transitions to verify that the oracle VLM’s annotations are well-aligned with our human annotations, with an average alignment score around 8, on a Likert scale from 1 to 10 (higher is better). We acknowledge that the Oracle may still have blind spots outside this holistic evaluation, but our downstream results demonstrate that DynaView still learns usable transition dynamics from these annotations.
>
> **(W2)** The reviewer concerns that our evaluation of the downstream visual narrative generation relies heavily on automatic VLM judges, with limited human analysis.
>
> We conduct a head-to-head human evaluation between DynaVieW (SFT) and BAGEL (SFT). We ask human testers to read the input textual narratives, and then compare storyboards generated by the two models (anonymously presented and randomly ordered). In total, we collected 400 preference votes. The results show that human testers prefer DynaVieW in 47.6% of cases, compared to 35.4% for BAGEL, with 17% of comparisons resulting in a tie. The human evaluation results align well with automatic VLM judges.
>
> **(W3 and Q4)** The reviewer asks the compute and latency costs of our model’s visual sequence generation, and how they scale to long state sequences.
>
> We have tested the compute and latency costs of models on the downstream visual narrative generation task using a single NVIDIA H100 GPU, averaging results over 300 random testing samples in VinaBench dataset. The latency of generating a visual narrative state (in seconds) with respect to the number of its preceding (previously generated) states in the context is shown in the following table.
>
> |# Preceding States|DynaVieW|BAGEL|
> |-|-|-|
> |1|407.71±19.71|406.77±20.66|
> |2|502.91±28.75|501.38±30.10|
> |3|582.73±35.10|580.32±35.30|
> |4|646.08±49.59|645.85±48.35|
> |5|724.30±61.51|722.40±62.08|
>
> The inference latency cost is almost linear with the state sequence length. Similarly, we also find the same trend for the compute cost. We can roughly estimate the compute and latency costs when we scale to long sequences, e.g., costs on visual narratives with 50-100 states will be roughly 10 times of the costs on visual narratives with 5-10 states.
>
> **(Q1)** The reviewer asks how the granularity of our transition schema and the schema token re-weighted loss would affect the performance of our DynaVieW model.
>
> We conduct ablation study on: (a) removing the schema token re-weighting of CE loss (**w/o Re-weighted Loss**), and (b) using a more coarse-grained transition schema without describing the fine-grained low-level visual transformations and their contributions (**w/o Low-level Trans.**).
>
> We compare the performances of our original DynaVieW and the ablated models on the downstream VinaBench dataset, under the zero-shot generalization setting used in Table 2.
>
> |Model|Non-Char. Ent.|Char. Num.|Char. Attr.|Time|Location (Align.)|Style|Character|Location (Cons.)|Average|
> |-|-|-|-|-|-|-|-|-|-|
> |DynaVieW|0.600|0.323|0.521|0.551|0.501|0.897|0.550|0.835|0.630|
> |w/o Re-weighted Loss|-0.005|-0.011|-0.005|-0.007|-0.003|-0.024|+0.019|+0.003|-0.004|
> |w/o Low-level Trans.|-0.047|0|-0.045|-0.024|-0.030|-0.203|-0.074|-0.026|-0.065|
>
> Without the schema token re-weighted CE loss, the model underperforms DynaVieW, although with a small performance margin. Fine-grained hierarchical transition descriptions are also necessary for the success of DynaVieW training. Using coarse-grained JSON transitions results in a decrease of 0.065.
>
> **(Q3)** The reviewer asks whether our DynaVieW model can perform inferences with free-form prompts instead of JSON-schema prompts of transitions?
>
> Yes, as shown in Table 3, our model achieves comparable performances of visual sequence (narrative) generation when prompted with transitions in natural language (NL) and JSON schema, where the natural language transitions are close to free-form prompts.
>
> **(Q5)** The reviewer asks whether our pre-trained DynaVieW model can achieve improvements on long-horizon videos’ state modeling outside the pretraining data sources?
>
> We clarify that VinaBench and LEGO samples are sourced from various long (hourly-level length) movies and ego-centric videos, respectively, both beyond the coverage of our pretraining data sources. Our zero-shot generalization results on these datasets in Table 2 and 4 verify that our pre-trained model can improve the modeling of visual state sequences from these out-of-distribution long-horizon videos.

---

### Official Review · Reviewer_RZuu · 2026-03-15

**Soundness:** 3
**Presentation:** 3
**Significance:** 2
**Originality:** 3
**Overall Recommendation:** 4
**Confidence:** 3

**Summary:**

This paper focuses on enabling multimodal models to better understand and simulate visual dynamics in real-world activities. MLLM often struggle to capture structured dynamic changes in the world, such as object interactions. And the issue in this disadvantage leads to underperfomrance in various tasks including narrative generfation, simulation. This paper proposed DynaVieW, a schema-guided world model designed to learn hierarchical visual dynamics through structured state-transition modeling. The key idea is to represent visual sequences as alternating states and transitions. 1. images extracted from video keyframes representing the world at a particular moment. 2. structured textual descriptions (in a JSON schema) that explain how the world evolves between adjacent states. To train the model, the authors construct state-transition sequences from videos using datasets such as Ego4D, AgiBotWorld, and ShareGPT4Video. Video keyframes are extracted as states, and transitions. The model is evaluted on visual narrative generation (VinaBench) and instruction-following world simulation (LEGO benchmark).

**Compliance With Llm Reviewing Policy:**

Affirmed.

**Final Justification:**

The author resolved some of my concerns like table 1 results. I keep my positive score.

**Key Questions For Authors:**

1. For  Depicted Transformations, why the performance is even better than Gold in Table 1.

**Limitations:**

They discussed that in the paper.

**Strengths And Weaknesses:**

Strength:

1. This paper has clear motivation and problem framing. I do believe the targetting issue is one of the key issues in MLLM.
2. The model jointly learns to predict structured transitions generate the next visual state. This unified framework aligns well with the goal of building general world models for multimodal AI.
3. The authors construct a substantial training dataset by extracting video keyframes and generating structured transition annotations using a VLM. This pipeline enables learning richer temporal dynamics than typical image-text datasets.

Weakness:

1. Using models like Gemma in this paper to predict the states to evaluate is not validated. Understanding dynamics is challenge for MLLM and it is hard to get a very credible results.
2. No ablation study to validate each component. For example, what is the usage of Multimodal Selective Attention in experiment.

---

> ### Author Rebuttal · Authors · 2026-03-30
>
> We thank the reviewer for acknowledging that our studied research problem is important in the field of multimodal LLMs, and our proposed method is well-aligned with the goal of world modeling in this field, enabling the learning of richer visual temporal dynamics.
>
> **(W1)** The reviewer concerns that in our controllability experiment in Table 3, using Gemma to predict the states for evaluation may not lead to credible results, because the model’s understanding of visual dynamics is not validated.
>
> We first clarify that we use Gemma-3-27B to generate only the JSON-schema transitions (instead of the visual states themselves) as potential input conditions for evaluating the controllability of simulating the next visual states. We also clarify that the next visual state in this experiment is not set to be fixed or specified, so there is no gold reference of what transition to the state should be generated, making the validation of Gemma predictions open and indeterminate.
>
> Indeed, Gemma-3-27B has not been broadly pre-trained on world modelling tasks to understand the visual dynamics, but it has strong instruction-following performance. Therefore, we enhance its capability of visual dynamics understanding through sophisticated prompts, to fine-grainedly instruct its transition prediction. Moreover, we expect Gemma to create diverse (more important than accurate in controllability experiments) transitions to test our DynaVieW model’s controllability of state simulation based on varied possible transitions, and noisy transitions can be used to examine the robustness (reflected by the low variance of scores shown in Table 3) of our model in resisting potentially infeasible state simulation conditions.
>
> **(W2)** The reviewer concerns ablation study to validate the contributions of each component in our proposed method.
>
> We conduct a detailed ablation study on each major component of our method, including: (a) removing the multimodal selective attention (**w/o Selective Attn.**), (b) removing the schema token re-weighting of CE loss (**w/o Re-weighted Loss**), (c) using a heuristic template to translate the JSON-schema transitions into natural language descriptions (**w/o JSON-Schema**).
>
> We compare the performances of our original DynaVieW and the ablated models on the downstream VinaBench dataset, under the zero-shot generalization setting used in Table 2.
>
> |Model|Non-Char. Ent.|Char. Num.|Char. Attr.|Time|Location (Align.)|Style|Character|Location (Cons.)|Average|
> |-|-|-|-|-|-|-|-|-|-|
> |DynaVieW|0.600|0.323|0.521|0.551|0.501|0.897|0.550|0.835|0.630|
> |w/o Selective Attn.|-0.018|-0.021|-0.019|-0.020|-0.016|-0.023|-0.007|+0.011|-0.013|
> |w/o Re-weighted Loss|-0.005|-0.011|-0.005|-0.007|-0.003|-0.024|+0.019|+0.003|-0.004|
> |w/o JSON-Schema|-0.019|-0.009|-0.013|-0.009|-0.010|-0.067|-0.031|-0.029|-0.028|
>
> Results above show that removing the multimodal selective attention consistently makes the model perform worse. Without the schema token re-weighted CE loss, the model also underperforms DynaVieW, although with a smaller performance margin. Our ablation on the schema further shows that JSON-structured descriptions of visual dynamics are necessary for the success of DynaVieW training. We will include these ablation study results in our camera-ready version.
>
> **(Q1)** The reviewer asks that in our validation results of the next state simulation in Table 1, why the depicted transformations of DynaVieW’s simulated state achieve an even better score (on Accept rate) than the gold reference state.
>
> We clarify that due to the open-ended nature of our state-transition modeling task, the gold reference state may not be the only possible next state continuation of the visual sequence. Therefore, at the validation phase, DynaVieW could simulate a next state that is different from the gold reference but still a possible continuation. Compared to the gold state, this simulated state could have more transformations depicted (i.e., lower None rate), including more reasonable transformations (i.e., higher Accept rate) but meanwhile also more unreasonable transformations (i.e., higher Reject rate) in trade-off.

---

### Decision · Program_Chairs · 2026-04-30

**Decision:**

Accept (regular)

**Comment:**

This paper proposes DynaVieW, a schema-guided multimodal world model for learning structured visual dynamics. Reviewers consistently agree that the work is well-motivated, technically sound, and addresses an important problem in multimodal world modeling . The proposed state–transition formulation and schema-aware design are considered meaningful contributions, and the empirical results demonstrate clear improvements over prior methods. The rebuttal further strengthens the paper by providing additional ablations, baseline comparisons, and human evaluation.

Overall, there is strong consensus among reviewers (all Weak Accept) that this is a solid and promising piece of work with clear potential impact. I therefore recommend Weak Accept.